# Early treatment regimens achieve sustained virologic remission in infant macaques infected with SIV at birth

Xiaolei Wang [1], Eunice Vincent[1], Summer Siddiqui[1], Katherine Turnbull [1], Hong Lu[2], Robert Blair [1], Xueling Wu [2], Meagan Watkins[1], Widade Ziani[1], Jiasheng Shao[1], Lara A. Doyle-Meyers [1], Kasi E. Russell-Lodrigue [1], Rudolf P. Bohm[1], Ronald S. Veazey [1] & Huanbin Xu [1] ✉

Early antiretroviral therapy (ART) in HIV-infected infants generally fails to achieve a sustained state of ART-free virologic remission, even after years of treatment. Our studies show that viral reservoir seeding is different in neonatal macaques intravenously exposed to SIV at birth, in contrast to adults. Furthermore, one month of ART including an integrase inhibitor, initiated at day 3, but not day 4 or 5 post infection, efficiently and rapidly suppresses viremia to undetectable levels. Intervention initiated at day 3 post infection and continued for 9 months achieves a sustained virologic remission in 4 of 5 infants. Collectively, an early intervention strategy within a key timeframe and regimen may result in viral remission or successful post-exposure prophylaxis for neonatal SIV infection, which may be clinically relevant for optimizing treatment strategies for HIV-infected or exposed infants.

Although the Berlin, London, and Düsseldorf adult patients appeared to have experienced a sterilizing human immunodeficiency virus (HIV) cure[1], the treatment regimen involved toxic chemotherapy followed by hematopoietic stem cell transplantation, which is not feasible for most patients due to the high risk and expense. In HIV-infected adult patients, proviral reservoirs are extremely stable during long-term antiretroviral therapy (ART), and become the driving force for viral rebound after analytic treatment interruption (ATI)[2–6]. Although the timing of viral genome integration is uncertain in acute HIV infection, adult non-human primate models of simian immunodeficiency virus (SIV) suggest that proviral reservoirs are established as early as 3 days post infection[7,8]. Mother-to-child HIV transmission can occur during in utero at late pregnancy, at the time of delivery or in the days or weeks after birth through breastfeeding[9,10]. Current interventions to prevent in utero HIV infection have been highly successful, yet ~150,000 children are still postnatally infected with HIV every year[11]. Although early initiation of ART in infants may reduce the viral reservoir size[12,13], there are no genuine cases of a sustained state of ART-free virologic remission or cure in pediatric AIDS clinical trials or case reports[12,14–19]. Note

that the presence of HIV reservoirs in tissues of human infants at or near the initiation of ART is rarely confirmed due to sample collection limitations. Moreover, current pediatric treatment regimens (e.g., dual nucleoside reverse transcriptase inhibitors/NRTI AZT and 3TC, plus protease inhibitors) lack integrase inhibitor, which may be unable to block viral genome integration in cells, likely predisposing to new or continual proviral reservoir seeding. In addition, recent studies show that infant macaques on late treatment with regimens containing integrase inhibitors also exhibited viral rebound after short-term ATI[20,21]. All of these suggest that proviral reservoirs, once established, are likely difficult to eradicate by conventional strategies and prevention of initial viral integration may be a key to overcome this major obstacle towards a pediatric HIV cure.

Newborn infants possess a unique immune system, characterized by the immature systemic immune system and the functional/mature mucosal system[22,23]. Neonates and infants infected with HIV develop disease rapidly, which is accompanied by compromised immune development and function[24,25]. We speculate that proviral reservoir seeding in infants exposed to HIV may be distinct from adults that have

[1]Tulane National Primate Research Center, Tulane University School of Medicine, 18703 Three Rivers Road, Covington, LA 70433, USA. [2]Aaron Diamond AIDS Research Center, Columbia University Vagelos College of Physicians and Surgeons, New York, NY 10032, USA. ✉e-mail: hxu@tulane.edu

larger numbers of potential target cells in tissues[26]. However, HIV/SIV viral reservoir seeding in developing infants exposed to viruses remain unclear. In this study, we examined viral reservoirs in neonatal macaques inoculated with SIV at birth and then given cART including dolutegravir (DTG) initiated at 3-, 4- or 5 days post infection. The results showed that proviral DNA was primarily detected in one of five neonates examined at 3 days post infection. One month of treatment initiated at day 3, but not delayed 1 day beyond, rapidly and efficiently suppressed viral replication to undetectable levels. Remarkably, in an additional group, 9 months of treatment initiated at day 3 resulted in a sustained virologic remission in 4 of 5 infants, as indicated by undetectable viral load and CD4+ T-cell preservation after analytical treatment interruption, and absence of viral rebound by further CD8+ cell depletion in vivo. These findings suggest that early interventions with proper treatment regimens may achieve HIV remission for at least some pediatric HIV infections.

## Results

### Viral reservoir seeding in tissues of neonatal macaques intravenously infected with SIV at birth

By nested qPCR (Supplementary Fig. 1a), 1 or 5 copies of SIV plasmid DNA, spiked with peripheral blood mononuclear cell (PBMC) genomic DNA ($10^5$ cell equivalents) from SIV naive macaques, could be consistently detected, as shown that ~1.57 or ~5.56 copies were determined, respectively (Supplementary Fig. 1b). Human CEM×174/SIV cell line (clone 3D8) is reported to contain a single copy of SIV provirus and is used as a standard to quantify proviral SIV DNA with Alu 1–2 primers. To increase the detection rate of integrated SIV DNA in rhesus macaque samples, additional rhesus Alu 3–4 primers were combined with Alu 1–2 primers in the current Alu qPCR assay (Supplementary Table 1). The data indicated that 5 copies of integrated SIV DNA in cell number equivalents could be efficiently quantified by nested Alu 1–4 qPCR, as shown by the value ($4.6 \pm 0.86$) which was close to the input. It appeared that HUT78/SIV cell line contained an average of 1–2 copies of proviruses per cell. This improved assay yielded higher levels of integrated proviral DNA in samples from chronically SIV-infected macaques when compared with conventional Alu 1–2 qPCR (Supplementary Fig. 1c, d). Together, improved nested (Alu) qPCR assay can increase sensitivity and detection rate of low copy numbers of total and proviral DNA. The values in this study were determined by a common nominal assay.

To validate and evaluate the effects of integrase inhibitor on viral genome integration in macaque cells ex vivo, proviral SIV DNA was measurement by nested Alu qPCR to compare virus-producing ability of early unintegrated SIV DNA in infant PBMC samples ($n = 3$) treated with integrase inhibitor. Cells were activated and then infected with SIV in the presence or absence of the integrase inhibitor raltegravir (RTG). The results showed that CA proviral DNA was not detected in PBMCs in presence of RTG, yet CA SIV RNA/DNA were detectable as early as 6 hours after SIV incubation. Most importantly, supernatant viral load levels had no significant difference at early SIV infection (3 and 7dpi) between RTG-treated and untreated cell samples (Supplementary Fig. 2a, b). This data suggest that viral genome integration might be blocked by RTG treatment while total SIV DNA likely represents an unintegrated form, yet the nested Alu qPCR assay reflects levels of genome integration in this assay. The early virus particles in cell supernatants are not likely produced from integrated SIV DNA in presence of RTG but may be produced from early unintegrated SIV DNA.

To determine viral reservoir seeding in infants exposed to HIV, we used the pediatric SIV macaque model to assess early events at precise timepoints representing HIV exposed neonates from maternal blood during birth and initial breastfeeding. Newborn macaques were intravenously inoculated with identical doses of SIVmac251 within 6 hours of birth, and euthanized at day 1, 2, 3, 5, and 7 post SIV inoculation for complete tissue collections (Fig. 1a). Due to limited volume of blood sampling for plasma viral load measurement, additional blood collections were staggered at some early timepoints post SIV infection and from other untreated SIV-infected infant cohorts that were euthanized at 28dpi or before treatment (Supplementary Table 2). The results showed that plasma virus was detected in ~50% neonates at 1 dpi and was positive in all infants thereafter (Fig. 1b), largely consistent with the dynamics of cell-associated (CA) SIV RNA in PBMC, axillary lymph node (LN) and colon (Fig. 1c–e). Furthermore, total SIV DNA were detectable in these systemic and mucosal lymphoid tissues of neonates as early as 1 dpi once exposed to SIV (Fig. 1f–h). Notably however, it appeared that integrated proviral DNA was undetectable until day 3 of infection, when very low levels of proviral DNA was first detected in axillary LN and tonsil tissues only from one in five animals. Strikingly, all neonates at 5 and 7dpi showed readily detectable CA proviral DNA in tissues (Fig. 1i–m). These findings indicate that irreversable reservoir seeding may not be fully established in neonates at 3 days post exposure, which may provide a critical window to block the establishment of proviral reservoirs for sustained viral remission or even a cure in pediatric HIV infection by appropriate early treatment interventions.

### Viral dynamics in neonates after treatment intervention initiated at day 3, 4, or 5 post SIV infection

To further compare the impact of early interventions on viral replication and viral reservoir eradication in infants infected with SIV, short-term cART (tenofovir/emtricitabine/DTG; TFV/FTC/DTG) was initiated at day 3, 4, or 5 after IV inoculation and continued for 28 days ($n = 3$ per group), compared with untreated infants ($n = 3$). In untreated infants, viremia plateaued at 7–9 dpi, then maintained a high set point. In comparison, interventions initiated at 3 dpi completely suppressed viral replication to undetectable levels by 7dpi and after, despite viral blips observed in some infants at 5 dpi (Fig. 2a). Unexpectedly, initiating treatment at day 3 post infection, compared with even one- or two-day delayed initiation (at 4 or 5 dpi), exhibited distinct outcomes in suppression of viral replication (Fig. 2a–c). Compared with treatment at 3 dpi, delayed initiation of treatment consistently resulted in persistent viral replication as shown by higher viremia at 7 and/or 14dpi in infants when cART was initiated at 4- or 5-days post infection (Fig. 2a–d). At day 28, levels of CA SIV RNA and total viral DNA significantly decreased, and integrated SIV DNA was essentially undetectable in PBMC, axillary LN and rectum of infants when cART was initiated at 3 dpi (Fig. 2e–g). However, CA SIV RNA in these tissues appeared to be accumulating when infant cohorts were treated at 5 dpi (Fig. 2e), likely reflecting the failure of virus packaging under treatment. Although treatment initiated at 4 or 5 dpi also caused decay of proviral reservoirs, total viral DNA was still relatively stable at 28dpi (Fig. 2f, g). These data suggest that timing of early intervention may be key to containing virus replication and preventing integration in infants exposed to HIV/SIV.

### Efficacy of intervention initiated at 3 days post infection on achieving sustained virologic remission in infants

To investigate outcomes of early interventions with prolonged cART in infants, an additional five newborn macaques were infected with SIV and treated daily with cART beginning at 3 dpi for 9 months. Consistent with Fig. 1b, plasma viral load was detected in these infants prior to early treatment. cART initiation at 3 dpi effectively suppressed viral replication in infants at scheduled timepoints throughout the 9 months of treatment, as shown by undetectable viremia and preserved CD4+ T cells thereafter under ART. After analytical treatment interruption, viral rebound was not detected in 4 infants (up to 3 months for two infants, 18 months for another two animals). However, one infant (NG24) exhibited viral rebound at 2 months after ATI, concomitant with CD4+ T-cell depletion at this timepoint (Fig. 3a, b,

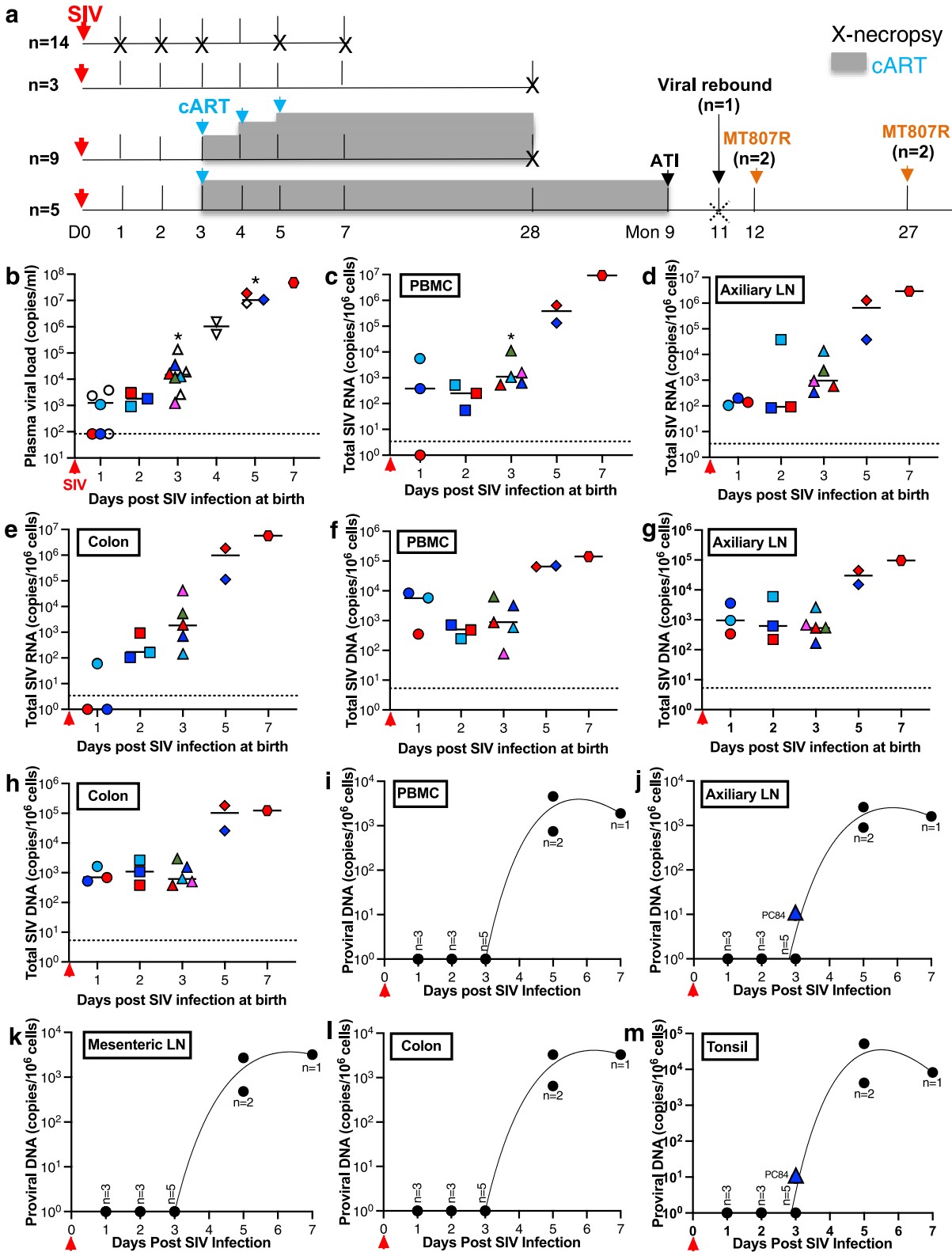

and gate strategy in Supplementary Fig. 3). Even so, viral rebound in NG24 was significantly delayed, compared to the viral recrudescence in SIV-infected adult macaques, which largely occurs between 10 and 21dpi when the same cART is initiated at 3 dpi[7,8]. Together, these data demonstrate that pediatric HIV remission is possible in a proportion of postnatally HIV-infected infants by early interventions.

To track viral reservoirs in these five infant macaques before and after ATI, we measured CA total SIV RNA/DNA and proviral DNA in PBMCs and axillary LN biopsies, which were collected at day 3, month 1, 4, 9, and 11. The results indicated that CA SIV RNA from both PBMCs and LNs was detectable at day 3 and month 1 post infection when cART was initiated at 3 dpi, and gradually diminished from months 4–9 after treatment (Fig. 3c, f). Further, although total SIV DNA was detected in tissues at 3 dpi, early intervention initiated at day 3 of infection could efficiently eliminate unintegrated SIV DNA from PBMCs and LNs by day 28 and beyond (Fig. 3d, g). Most importantly, proviral DNA was not

**Fig. 1 | Early viral reservoirs in systemic and lymphoid tissues in neonatal macaques intravenously infected with SIV at birth. a** Outline for infant macaque sampling and early combined retroviral therapy (cART) in the study. **b** Plasma viral load (PVL) in blood at necropsy at day 1 ($n = 3$), 2 ($n = 3$), 3 ($n = 5$), 5 ($n = 2$), or 7 ($n = 1$) and staggered blood samples at day 1 ($n = 3$), 3 ($n = 3$), or 5 ($n = 1$) from untreated individual neonates after early SIV infection (animals shown in Supplementary Table 2). Data are presented as the median PVL in blood samples from individual infant macaques. *$p$ value for the comparison of median value of PVLs with 1 dpi are <0.05, which was determined by two-tailed $t$ test. Colored symbols represent individual neonatal animals that were euthanized. Opened black symbols represent plasma viral load from staggered blood samples collected from untreated infant animals, who were euthanized at day 28 post SIV infection. **c–h** Total cell-associated SIV RNA (**c–e**) or SIV DNA (**f–h**) in PBMC, axillary LN and colon, collected from neonates euthanized at day 1 ($n = 3$), 2 ($n = 3$), 3 ($n = 5$), 5 ($n = 2$) or 7 ($n = 1$). Data are presented as scatter plot from individual infant macaques, with median value. Statistical significances were analyzed by a Mann–Whitney test. *$p < 0.05$, compared with 2dpi. **i–m** Detectable integrated SIV DNA with the days post SIV infection in PBMC, auxiliary LN, mesenteric LN, colon and tonsil in neonates. Tissue lymphocytes were collected from euthanized neonatal macaques at 1 dpi ($n = 3$), 2 dpi ($n = 3$), 3 dpi ($n = 5$), 5 dpi ($n = 2$) and 7dpi ($n = 1$). Cell-associated proviral DNA were measured by Alu-based nested qPCR. Data were analyzed by curve fitting using nonlinear regression. Source data are provided as a Source Data file.

detected in four infants, but one infant animal (NG24) had very low levels of integrated SIV DNA in axillary LN at multiple timepoints throughout treatment (Fig. 3e, h). It remains unknown if these proviral reservoirs in axillary LN or/and other tissues could contribute to viral rebound and dissemination after analytical treatment interruption in this animal. These data were also consistent with in situ hybridization (RNAScope) analysis, as shown by detectable SIV RNA in both LN and rectal tissues of infant NG24 at 2 months post ATI, compared with undetectable SIV RNA in both tissues of the other four infants at this timepoint (Fig. 3i–j).

Given distinct outcomes of infants on 9 months early cART, in which four infants showed viral remission while one showed viral rebound off-ART, the potential correlates of immunity were further investigated. First, all infants studied were negative for Mamu-A*01, Mamu-B*08 and Mamu-B*017 (Supplementary Table 3), which are associated with elite control in some SIV-infected rhesus macaques[27]. Second, neither anti-SIV gp140 Abs nor autologous neutralizing Abs against SIV Env were elicited in these SIV-infected infant macaques, except NG24 which displayed detectable anti-SIV gp140 Abs 2 months after ATI (Fig. 3k, l). To prove whether a viral remission or sterilizing cure was achieved, CD8+ cells were depleted in the four virus-controlling infants treated by rhesus anti-CD8α Ab in vivo as previously described[28,29]. As shown in Fig. 3m, CD8+ cells were completely depleted for up to 3 months in these infants after M-T807 Ab administration in vivo. Notably, no viral rebound or blips were detected during CD8+ cell depletion in these four infants. Combined, these findings demonstrated that pediatric viral remission was achieved in these SIV-infected infants when early intervention combined with integrase inhibitor was initiated at day 3 post infection, and did not correlate with known protective MHC alleles, neutralizing Ab, or CD8 T-cell responses.

## Discussion

Utilizing the pediatric SIV macaque model of HIV, our findings provide key information and insight into early interventions in pediatric HIV infection including: (1) dynamics of tissue viral reservoir seeding in neonates in early stages of SIV infection; (2) rapid suppression of viral replication in infants by early ART combined with an integrase inhibitor; and (3) sustained virologic remission achieved by early treatment regimens. In summary, ART-free viral remission (or perhaps successful post-exposure prophylaxis) may be achievable in a proportion of HIV-infected infants by rational early treatment regimens.

Given the unique immune system of developing neonatal infants, levels of immune maturation, viral susceptibility, and viral reservoir seeding in neonates exposed to HIV likely differs from that of adults[24,25]. Although unintegrated viral DNA is unstable, it is believed to be capable of generating productive virus particles in early HIV/SIV infection[30–35] even in the presence of integrase inhibitors[34,35]. As shown in Supplementary Fig. 2, RTG essentially blocks viral genome integration in PBMCs infected with SIV in vitro, while unintegrated SIV DNA and supernatant viruses in the presence of RTG were still detectable at 3 and 7dpi, suggesting that early

unintegrated viral DNA per se potentially generates virions, which is consistent with previous reports[30,34,35], and could explain the early emergence of viremia in neonates in vivo within 3 days of exposure. On the other hand, HIV genome integration is believed to be an essential hallmark for stable HIV replication and persistence, as integrated DNA is the dominant template for persistent HIV replication[36], and difficult to eradicate by conventional cure strategies[37–39]. Insight into the timing and onset of viral reservoir seeding in infants exposed to HIV is critical for providing rationales for pediatric HIV therapy. Current scalable assays to measure viral reservoirs might not truly reflect the bona fide functional viral reservoir size in tissues[40–48]. For example, a small proportion of proviruses might persist in various tissues during long-term treatment that may not be detected in peripheral blood and LN samples that are generally examined. Viral reservoirs examined by Quantitative Viral Outgrowth Assays (QVOA) or Tat/rev Induced Limiting Dilution Assays may not fully be reactivated by stimulators, especially when the proviruses are located in the gene desert sites of chromosomes enriched in repressive chromatin marks[45,49–51]. Near full-length individual proviral sequencing and intact proviral DNA assay (IPDA) also may not accurately distinguish the functional viral reservoir, as indicated by existence of intact proviral quasispecies and viral polymorphism[52,53], the difficulties inherent in predicting the production of replication-competent virions in the viral life cycle of HIV/SIV, questionable functionality of viral genomes by various Indels and mutations[47,54–56], or potential functional revertant mutations from deficient viral genomes[57,58]. Therefore, there is no gold standard approach to assess the genuine size of viral reservoirs in the body because of limited tissue sampling, and sensitivity limitations[59]. Although qPCR may overestimate the viral reservoirs because of defective proviruses, total proviral DNA levels can still estimate the size of the intact viral genome, which accounts for ~11.7% in HIV+ patients and even higher fractions in SIV+ macaques on cART[60,61]. Here we combined a ddPCR-based total viral RNA assay with nested qPCR for both total SIV DNA and integrated SIV DNA quantification, the latter targeting multiple repetitive Alu DNA islands (6–13% of genomic DNA) in macaque chromosomes, providing a rapid approach to generate comparable levels of both total SIV DNA and integrated DNA for each sample. With our improved Alu qPCR assay based on previous reports[8,21,62], our in vitro data indicated that CA proviral DNA was not detected in RTG-treated PBMC infected with SIV, compared with untreated controls, albeit considerable total viral DNA was still detectable. Note that integrated proviral SIV DNA before 5 dpi (not examined at 4dpi) was not detected in infant PBMC infected with SIV in vitro in absence of RTG treatment. These results were de facto consistent with in vivo data that viral genome integration was not observed in PBMCs in neonates within 1–3 dpi after SIV infection. Likewise, our in vivo data showed that total CA SIV DNA/RNA was detected in systemic and lymphoid tissues as early as 1 dpi in neonates once infected with SIV at birth, suggesting that detectable viremia in very early SIV infection is not due to residual virus from the inoculation, but rather, due to rapid viral replication. Viral genome integration appeared to show distinct patterns in very

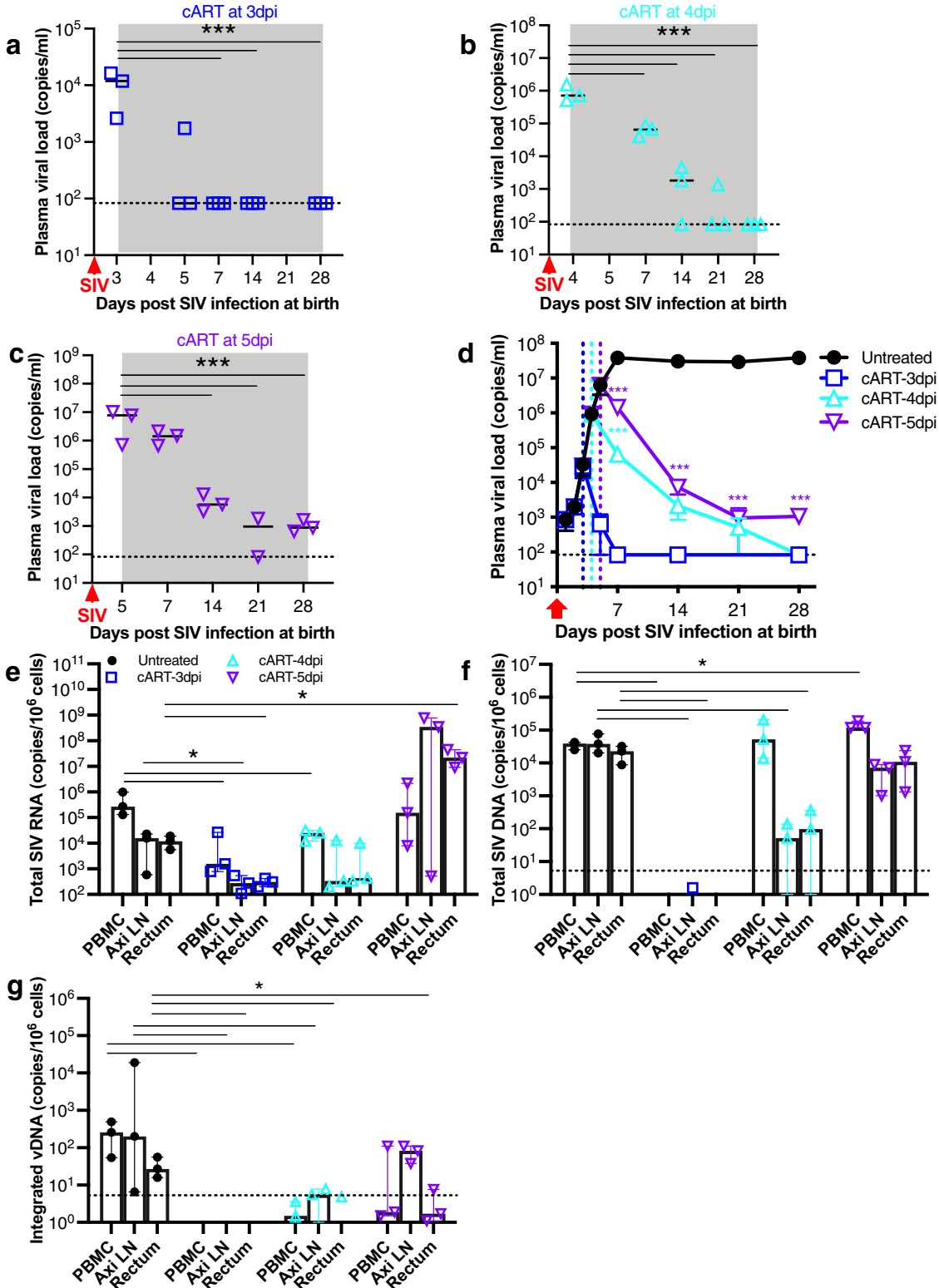

early SIV infection; however, we cannot rule out the possibility that low levels of integrated or proviral SIV DNA may not be detected in very early infection due to assay sensitivity limitations, inability to examine all tissues in the body, and the potential lack of chromosomal Alu islands situated near the integrated SIV genome in some target cells. In this study, we examined multiple tissues, including spleen, thymus, jejunum, colon, and liver, yet proviral DNA was normally not detected in these tissues in very early SIV infection unless otherwise reported. Nonetheless, further insight into the

dynamics of initial reservoir seeding, and how this may be impacted by different early treatment regimens to improve clinical outcomes will require further animal studies, testing different combinations of antiretroviral drugs and optimizing the timing of treatment initiation.

All children exposed to HIV before, during, and after birth should receive early antiretroviral (ARV) drugs to reduce the risk of HIV acquisition, morbidity, and mortality[12,16,25,63,64]. Although incorporation of nucleoside reverse transcriptase inhibitors (NRTIs) into

**Fig. 2 | Suppression of viral replication in SIV-infected infant macaques on short-term early interventions initiated at 3-, 4-, or 5-days post infection.** Newborn macaques were intravenously inoculated with SIVmac251 within 6 hours after birth, followed by combined retroviral therapy (cART) with initiated at 3 dpi (**a**, *n* = 3), 4 dpi (**b**, *n* = 3), or 5 dpi (**c**, *n* = 3) till 28 days of age. Data are presented as the levels of plasma viral load in blood samples, with median value of individual infant macaques at different timepoints. ***p < 0.001, compared with the timepoint prior to treatment, determined with two-tailed *t* test. **d** Statistical analysis of plasma viral load in infant cohorts by early treatment initiated at −3 (*n* = 3), −4 (*n* = 3) or −5 dpi (*n* = 3) or without treatment (*n* = 3, combined with plasma viral load from staggered blood samples within 28 days of infection), in which all infant macaques were intravenously infected with SIV at birth and euthanized at 28dpi. Data are presented as the mean ± SEM of plasma viral load for each group at different timepoints. ***p < 0.001 for the comparison between 4 or 5 dpi with 3 dpi at the same timepoints. Statistical significances were evaluated by two-tailed *t* test. Colored lines represent individual neonatal animals that received early anti-retroviral drugs at 3, 4, or 5 dpi. The black line represents plasma viral load from staggered blood samples before treatment and collected from untreated infant animals. **e**−**g** Levels of total SIV RNA, total SIV DNA and integrated SIV DNA in lymphocytes freshly isolated from PBMC, axillary LN and rectum in infant macaques (*n* = 3 each group) at day 28 of age. Data are presented as box and scatter plot, median with 95% CI. *p < 0.05, compared with the untreated animal control for each tissue (*n* = 3), determined by two-tailed *t* test. Note that early interventions initiated at 3 dpi, compared to interventions with one or two days delayed (initiated at 4 or 5 dpi), conferred rapid and efficient suppression of viral replication. Source data are provided as a Source Data file.

the nascent viral DNA terminates its synthesis in the viral life cycle, current early ART regimen with a combination of dual-NRTI (AZT + 3TC) plus protease inhibitors (LPV/r) may not fully prevent initial viral RNA/DNA production, likely leading to potential proviral reservoir seeding in infants exposed to HIV. Abundant unintegrated viral DNA presents a high risk of viral genome integration in the absence of INSTI treatment. In such situations, the addition of integrase inhibitors might prevent potential proviral reservoir seeding in early pediatric HIV infection. Of the integrase inhibitors, DTG is a very well-tolerated, highly effective, and affordable INSTI drug for low- and middle-income countries[65] and achieves more rapid and complete viral suppression compared to Raltegravir[66]. Although one study in Botswana suggested DTG use in early pregnancy was associated with neural tube defects (NTDs) in newborns[67], further data from this cohort also indicated the risk was only slightly greater compared to non-DTG ART cohorts[68]. Furthermore, worldwide pharmacovigilance data do not provide a strong support for a relationship between NTDs and DTG[69,70] and is nonetheless irrelevant to the treatment of HIV-infected infants.

Early ART in infants may restrict viral reservoirs and reduce HIV proviral DNA[12,13,16,63,64,71–75]. However, there is no reported case of ART-free sustained virologic remission or a cure in HIV + infants treated by early or late initiated ART in pediatric AIDS clinical trials and cases of any age, including the Mississippi infant, Canadian, or South African Children with HIV Early Antiretroviral Therapy (CHER) trials, IMPAACT, Pediatric Early HAART and Strategic Treatment Interruption Study (PEHSS), Botswana and European Pregnancy and Pediatric HIV Cohort Collaboration (EPPICC) studies. Unfortunately eventual viral rebound has inevitably been observed during the follow-up of analytical treatment interruption[13–15,18,25,72,76–79]. The Mississippi baby, treated 30 hours after birth, still showed viral rebound even after 27-months ATI[78], although it remains unknown whether proviral reservoirs are established during pregnancy due to in utero HIV infection[10]. Furthermore, viral reservoirs are rarely examined in tissues of infants owing to limited sample collections at the initiation of ART. Also, conventional regimens administered to infants generally lack integrase inhibitors, likely resulting in proviral reservoir establishment in infants despite therapy. Building on primary patterns of viral reservoir seeding in infants exposed to SIV at birth, intervention initiated at day 3, but not later, can rapidly and efficiently suppress viral replication to undetectable levels. These results suggest that the timing of early intervention initiation is a critical determinant for ART-free virologic remission in pediatric HIV infection. Thus far, the 4 cured infants are still alive and healthy with complete ART-free viral remission. Early interventions in newborn macaques exposed to SIV even after showing initial viral infection may result in successful prophylaxis (cure?) or at least prevent against SIV replication and dissemination, as indicated by eventual viral reservoir clearance and no viral rebound in cases following ATI and CD8 + cell depletion in vivo. In comparison, these effects have not been achieved in HIV-infected human adults or SIV-infected macaque adults on early DTG-based ART[2,7,80,81], or have not

been observed in SIV-infected infants on late treatment[20,21]. Combined, the results indicate effective eradication (or prevention) of established viral reservoirs by rational early treatment interventions may be key for achieving ART-free viral remission in HIV-infected/exposed infants, since establishment of even a few viral reservoirs correlate with high risk of viral rebound after ATI[16,82–85].

Contradictory outcomes in adult macaques regarding early treatment and viral rebound after ATI are noted[7,8,81]; conceivably, the differences may lie in efficiency of viral reservoir clearance by deferring early antiretroviral drugs or/and host immunity. In SIV-infected macaques on 16-months therapy, there are still significantly higher frequencies of intact SIV genomes compared to ART-treated HIV-infected humans[61]. In fact, intact HIV genomes decline more rapidly within the first seven years, then more slowly afterwards, and the half-life of the reservoirs is ~4.0 years until year seven and ~18.7 years thereafter[4,5,86]. Also, residual HIV reservoirs still exist when cART is initiated during primary HIV infection in adults[87]. To date, few HIV-infected individuals remain healthy without treatment, in which long-term treatment pressure may drive proviruses to accumulate in human inactive gene regions of host chromosomes, likely stymieing new virus production[88,89]. It is reported that prophylactic broadly neutralizing antibody (bnAb) therapy given at 30 h or cART regimen at 48 h in one-month-old macaques after oral exposure to low pathogenic SHIVSF162P3[90,91] could clear tissue viral reservoirs[92,93], in which all animals show aviremic outcomes over time even at initiation of treatment. However, oral inoculation has different viral kinetics and viral reservoir seeding is unknown in these infants, although it is unlikely bnAb treatment could fully block intracellular virus replication, integration and production in SHIV-infected cells and completely eradicate productive/latent reservoirs in developing infants. However, these data are consistent with ours as this may also have contributed to viral reservoir clearance by very early treatment. Together, multiple factors, including viral reservoir seeding, timing of intervention, route of perinatal infection as well as pathogenicity of HIV/SIV/SHIV strains, may determine ultimate clinical outcomes in infants on early treatment.

The distinct viral reservoir clearance and discrepant outcomes of early intervention in infants likely evolves immune maturation, ontogeny, susceptibility, host genetics, or other maternal factors[25]. There are also tremendous gaps in our understanding of pediatric HIV infection and efficacy of early treatment, e.g., why some infants infected with HIV/SIV show rapid AIDS development but not others. HIV remission is proposed to be associated with multiple factors including initial viral suppression rate and rapidity of early ART initiation[94]. The latter is confirmed by this study that timing and regimen of early ART initiation likely determines the suppression of viral replication, decay of viral reservoirs and clinical outcomes. A small proportion of the HIV-infected infants develop de novo potent and broadly neutralizing antibodies (bnAbs) that respond to different and perhaps new epitopes of HIV-1 Env as early as one year post infection, compared with bnAb development in HIV + adults after 2-3 years of

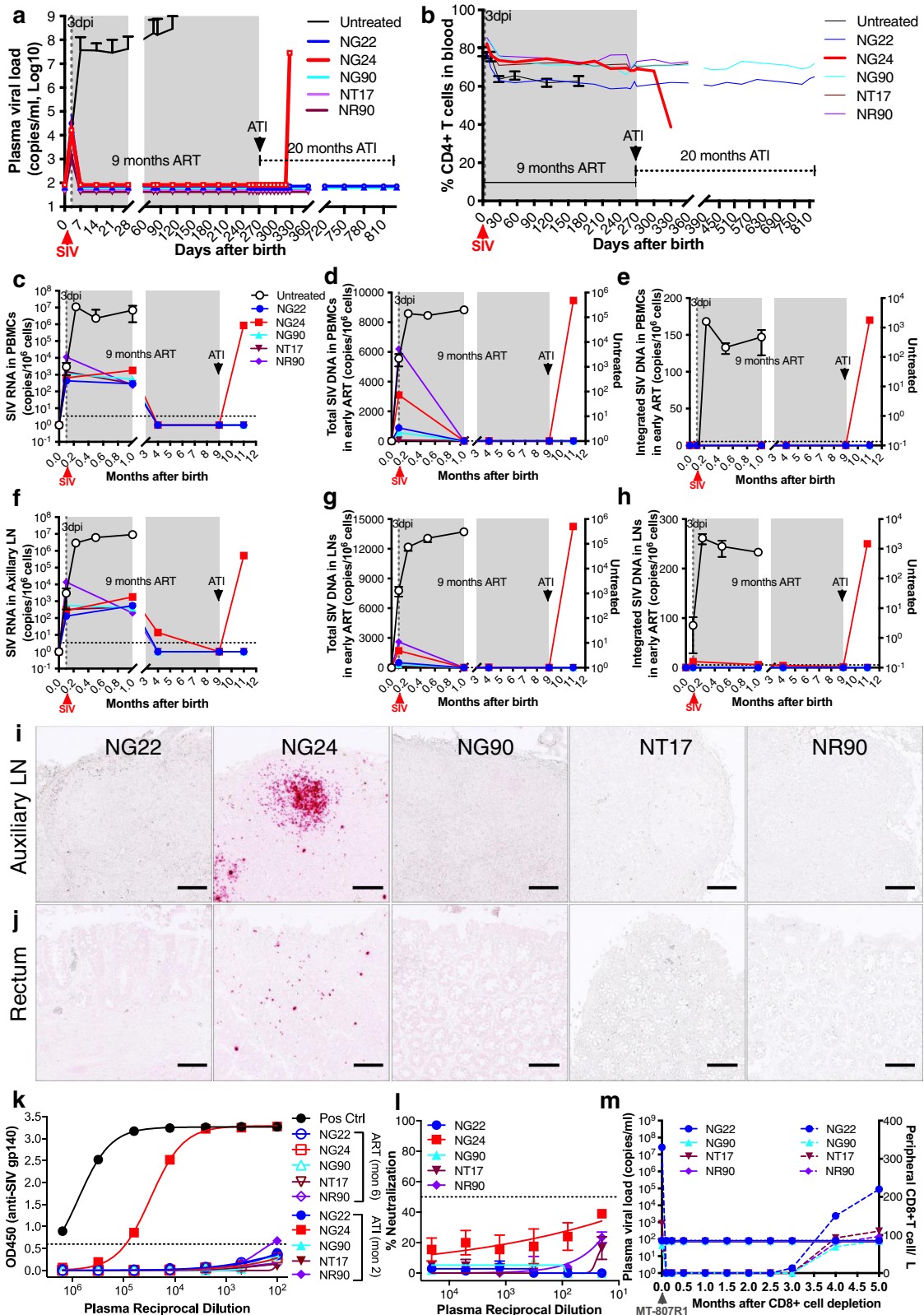

infection[95]. However, neutralizing Abs against SIV Env were not detected in any of five SIV-infected infant macaques on early ART, likely because SIV Env does not effectively elicit nAbs[96]. Interestingly, neither nAbs nor CD8 + cells were responsible for protection against SIV in the four infant macaques with viral remission, so the mechanism behind this discrepancy in proviral reservoir seeding of infants at the early stages of infection merits further investigation.

In summary, our study suggests that the timing and regimen of treatment intervention may determine the efficiency in viral reservoir clearance and clinical outcomes of pediatric HIV infection. These findings demonstrate that sustained HIV remission (or successful PEP) may be possible in infants postnatally exposed to HIV by certain early treatment regimens, providing a rational and translatable pediatric cure strategy for the clinic.

**Fig. 3 | Efficacy of 9 months early interventions on the sustained virologic remission in SIV-infected infant macaques when intervention is initiated at 3 days post infection.** Five newborn macaques were intravenously inoculated with identical SIVmac251 inoculum within 6 hours after birth, followed by antiretroviral therapy (ART) initiated at 3 dpi for 9 months, compared with age-matched infant macaques without treatment. **a** Plasma viral load in five individual SIV-infected infants on early intervention initiated at 3 dpi (colored lines, $n = 5$) or age-matched infant macaques without treatment (solid dark line, $n = 9$. Data are presented as the mean ± SEM of plasma viral load of untreated infant animals at designated time-points). Infant NG24 was euthanized at 2 months post analytical treatment interruption (ATI) when viral rebound was observed. Four infants on early intervention showed viral remission throughout the study (up to 18 months for two infants or 3 months for another two animals off-ART to date). **b** Changes in peripheral CD4+ T cells gated CD3+ T-cell populations in five individual SIV-infected infants on early intervention (colored line, $n = 5$), compared with those of age-matched SIV-infected and untreated (solid dark line, $n = 9$. Data are presented as the mean ± SEM of % CD4+ T cells of untreated infant animals at specific timepoints). Peripheral CD4+ T-cell loss was observed in infant NG24 at 2 months post ATI. Levels of cell-associated total SIV RNA, total viral DNA, and integrated SIV DNA in PBMCs (**c–e**) and axillary LN biopsies (**f–h**) from five individual infant macaques on early

intervention ($n = 5$) or untreated infant cohort post SIV infection (dark line, $n = 3$. Data are presented as the mean ± SEM of cell-associated SIV RNA/DNA or integrated proviral DNA in peripheral and LN tissues at designated timepoints). Colored lines represent individual neonatal animals, who were infected with SIV at birth, received early ART at 3 dpi and continued for 9 months, followed by ATI. The black line represents plasma viral load or cell-associated SIV RNA/DNA in age-matched infant animal cohorts without treatment. In situ SIV RNA + cells in axillary LN (**i**) and rectal (**j**) biopsies in five infant macaques at 2 months post ATI. SIV RNA was detected in both LN and rectal tissue of infant NG24 only. Bar, 100 μm. **k** Anti-SIV gp140-specific antibodies in the individual infants ($n = 5$) before and after ATI. Anti-SIV gp140 antibodies were detected in infant NG24 only at 2 months post ATI when viral rebound was observed. **l** Autologous neutralizing antibody against SIV Env in infants after 2 months of ATI (positive control/Pos Ctrl with black line included). Neutralizing antibodies were not elicited in these infants ($n = 5$). Data are presented as the mean ± SD of technical duplicates, representative of two independent experiments. **m** Absolute CD8+ T-cell numbers after anti-CD8α Ab administration and viral rebound after CD8+ cell depletion in vivo, in which plasma viral load or viral blips were not detected in the individual 4 infants up to 3 months to date. Source data are provided as a Source Data file.

## Methods

### Ethics statement

All animals in this study were housed at the Tulane National Primate Research Center in accordance with the Association for Assessment and Accreditation of Laboratory Animal Care International standards. All studies were reviewed and approved by the Tulane University Institutional Animal Care and Use Committee under protocol number P0401. Animal housing and studies were carried out in strict accordance with the recommendations in the Guide for the Care and Use of Laboratory Animals of the National Institutes of Health (NIH, AAALAC #000594) and with the recommendations of the Weatherall report: The Use of Non-Human Primates in Research. All clinical procedures were carried out under the direction of a laboratory animal veterinarian. All procedures were performed under anesthesia using ketamine, and all efforts were made to minimize stress, improve housing conditions, and to provide enrichment opportunities (e.g., objects to manipulate in cage, varied food supplements, foraging and task-oriented feeding methods, interaction with caregivers and research staff).

### Animals and virus

A total of 31 newborn, Indian-origin rhesus macaques (Macaca mulatta) with random sex difference were utilized in this study. Newborn macaques were intravenously inoculated with 100 TCID50 SIVmac251 on the day of birth (<6 h, day 0). To examine viral DNA integration in tissues, 14 neonatal macaques were euthanized for complete tissue collection at day 1 ($n = 3$), 2 ($n = 3$), 3 ($n = 5$), 5 ($n = 2$) and 7 ($n = 1$) post SIV inoculation. An additional 12 infant macaques received combined antiretroviral treatment (cART) with three anti-HIV drugs [tenofovir (TFV), 20 mg/kg/day; emtricitabine (FTC), 30 mg/kg/day; and DTG, at 2.5 mg/kg/day] initiated at day 3 ($n = 3$), 4 ($n = 3$), 5 ($n = 3$) or no treatment ($n = 3$) post SIV infection for 28 days. Staggered early blood sampling (1.2cc EDTA blood; newborns ~450 g) was collected from untreated animals or animals before early treatment. For a long-term intervention, five additional animals were infected and treated with cART initiated on day 3 and continued daily for 9 months. TFV and FTC were kindly provided by Gilead Inc. (Foster city, CA), and DTG was kindly provided by ViiV Healthcare (Research Triangle, NC). Blood was collected weekly to monthly, and LN and rectal biopsies were collected at 1 month on cART, and 2 months after ATI. Animals were euthanized for tissue collection if viral rebound after treatment interruption was observed. Plasma and single-cell suspensions were prepared to examine plasma viral load, CA SIV RNA/DNA, and for flow cytometry analysis. Previous plasma and PBMC data from age-matched SIV-

infected infant macaques ($n = 9$), which were infected with the same SIV lot, dose, and route were used to compare plasma viral load and the percentages of peripheral CD4+ T cells.

### Tissue collection and phenotyping

PBMC were isolated from EDTA-treated venous blood by density gradient centrifugation with Lymphocyte Separation Medium (Cat No: ICN50494, Fisher Scientific). Tissue lymphocytes were isolated from the LN and intestinal tissues. Briefly, LNs were minced, passed through nylon mesh screens, and washed twice in RPMI containing 5% fetal calf serum. Intestinal tissues were first cut into small pieces and incubated with RPMI containing EDTA for 30 minutes with shaking at 37 °C, followed by collagenase digestions (two 30 min intervals) and enrichment by Percoll density gradient centrifugation. Lymphocytes were processed for cellular DNA/RNA extraction or analyzed by flow cytometry as we previously reported[97]. Cells were stained with: CD3-Alexa Fluor 700 (clone SP34-2, Cat No: 557917, Lot No: 0293255), CD4-Brilliant Violet 711 (clone OKT4, Cat No: 317440, Lot No: B293041, BioLegend), CD8-APC-H7 (clone SK1, Cat No: 560179, Lot No: 1039448), CD8-PE (clone RPA-T8, Cat No: 555367), and LIVE/DEAD Fixable Aqua Dead Cell Stain Kit (Cat No: L34957, Lot No: 1836642, Invitrogen, Grand Island, NY) with final concentration of 50× dilution. Isotype-matched controls were included in all experiments. All antibodies and reagents were purchased from BD Biosciences Pharmingen (San Diego, CA) unless otherwise noted. Samples were resuspended in BD Stabilizing Fixative (BD Biosciences) and acquired on a FACS FORTESSA (Becton Dickinson, San Jose, CA). Data were analyzed with FlowJo software (Version 10.8.1, Tree Star, Ashland, OR).

### Viral genome integration in peripheral blood mononuclear cells infected with SIV and integrase inhibitor treatment in vitro

A total of $3 × 10^7$ PBMC from uninfected 1-month-old infant macaques ($n = 3$) were resuspended in complete media containing RPMI1640, 15% FBS, 100 U/mL pen/strep and 30 U/mL IL-2 and activated for 3 days by NHP T-Cell Activation/Expansion Kit (Cat No: 130-092-919, Miltenyi Biotec). Cells were washed and incubated with 100 TCID50 SIVmac251 for 2 hours in fresh media, then washed for 3 times to remove free viruses, and seeded as $1 × 10^6$/mL/well in the 24-well plate in absence or presence of raltegravir (RTG, Cat No: ARP-11680, NIH AIDS Reagent Program) at final 1 μM concentration. Cell pellets and supernatants were harvested before SIV infection, at hour 6, 8, 10, and 12 and the day 1, 2, 3, 5, and 7 after infection to measure viral load, CA SIV RNA, total SIV DNA and proviral DNA.

### Genomic DNA and total RNA extraction

Fresh PBMC or tissue lymphocytes (~$10^7$ in total per samples) were processed to extract cellular genomic DNA and cellular RNA with a AllPrep DNA/RNA Mini Kit (Cat No: 80311, Qiagen) according to the manufacturer's instructions. Viral RNA in plasma was directly isolated using the QIAamp Viral RNA Mini Kit (Cat No: 52962, Qiagen). The extracted cellular DNA and RNA samples were stored at −80 °C for qPCR analysis.

### Quantification of plasma viral load and CA SIV RNA and DNA

Specific primer sets and probes were synthesized by integrated DNA technologies and used to measure plasma viral load and CA SIV RNA/DNA, seen in Supplementary Table 1. Plasma viral load targeting SIV gag is used to determine plasma viral load with a limit of detection of 83 copies per mL. To quantify CA SIV RNA, the extracted RNA was reverse-transcribed into cDNA using a SuperScript III first-strand synthesis system (Cat No: 18080051, Invitrogen) in a thermocycler at 25.0 °C for 5 min and 50.0 °C for 60 min, followed by an enzyme inactivation step at 70.0 °C for 15 min. cDNA from cell-derived RNA was further used to quantify total SIV transcripts by digital droplet PCR (QX100 Droplet Digital qPCR system, Bio-Rad). Samples were run in duplicate in a 20 μL volume containing Supermix, 250 nM primers, 900 nM probe and 2 μL undiluted cDNA under the following cycling conditions: 10 min at 95 °C, 40 cycles of 94 °C/30 s and 63 °C/60 s, and a final step at 98 °C for 10 min. SIV gag primer/probe set was shown in Table S1. Droplets were analyzed by the QuantaSoft™ Software (Regulatory Edition #1864011, Bio-Rad) in absolute quantification mode. Samples that yielded <10 positive events in total 10,000 events were reported as undetectable SIV RNA. Copies of SIV RNA, expressed as copies per 1 million cells, were measured and normalized to cellular input, as determined by copies of genomic CCR5 (single copy rhesus macaque CCR5 DNA per cell)[81].

To quantify SIV DNA, nested qPCR was run in parallel to comparably quantify both total viral DNA and integrated proviral DNA. Briefly, the pre-amplification reactions were performed using SIV LTR U5 primer pairs (total SIV DNA and standard) or forward SIV LTR U5 primer combined with outward primers 1–4 targeting multiple Alu islands (integrated SIV DNA, Alu targets include conserved human/rhesus Alu DNA[62] and rhesus specific Alu sequences) on a 7900HT Sequence Detection System (Life Technologies). The primer/probe sets for total SIV DNA and integrated SIV DNA quantification were shown in Table S1. The reaction conditions were performed as follows: each 25 μL of reaction mix, contained 1× PCR buffer, 0.2 mM dNTPs, 2 mM $MgCl_2$, 0.8 μM of each primer and 0.5 U Taq DNA polymerase (Cat No: 10342046, Invitrogen Life Technologies) and was run for a 5-minute hot start at 95 °C, followed by 20 cycles of denaturation at 95 °C for 30 seconds, annealing at 63 °C for 30 seconds and extension at 72 °C for 3 minutes. Next, 2.5 μL of each amplicon was further amplified with same primer/probe set targeting SIV LTR U5 by real-time PCR reaction using 40 cycles at 95 °C for 15 seconds and 63 °C for 1 minute. Highly reproducible calibration curves were generated by plotting Ct values against log-transformed concentrations of internal standard. Standard curves were generated using known copy numbers of initial target plasmid standards diluted in cellular DNA from SIV naïve rhesus macaques. The calibration curves and internal regression curves were used for interpolating initial copies of each target in unknown samples. A non-template control and extracted cellular DNA from the HUT78/SIVmac239 cell line (positive control) were included in the qPCR reactions. Quantification of total SIV DNA and integrated proviral DNA, which were isolated from $10^7$ cell equivalents for individual samples, was expressed as copies per 1 million cells (LOD, 5.3 per one million cell equivalents, determined by at least ten replicates). Cell numbers were normalized by copies of genomic CCR5 DNA per cell (single copy rhesus macaque CCR5 DNA per cell[98,99]). Combined additional unique rhesus Alu repetitive sequences, optimized two-rounds PCR amplification (nested qPCR) potentially increase the sensitivity, specificity, and detection rate of total SIV DNA or integrated SIV DNA, which are validated by nested qPCR, as indicated by detection rate and values relative to the input: a) 1 or 5 copies of SIV plasmid spiked with genomic DNA of SIV naïve PBMCs; b) low copies of integrated SIV DNA in genomic DNA with cell number equivalents, including CEM×174 cell line (Cat No: ARP-13239, Clone 3D8) carrying single copy of proviral SIV DNA[100] and HUT78/SIV cell line[101] (Cat No: ARP-160) (NIH AIDS Reagent Program); and c) comparison of integrated SIV DNA levels in rhesus samples measured by nested qPCR using Alu 1–2 or Alu 1–4 primers (Supplementary Fig. 1).

### Detection of SIV RNA by in situ hybridization

To identify the SIV RNA in LN and rectal biopsies, formalin-fixed, paraffin-embedded sections (5-μm thickness) were cut and adhered to glass slides. After deparaffinization in xylene, and serial dehydration with 50%, 70% and 100% ethanol, sections were subjected to an RNAscope assay with RNAscope2.5 HD Detection Reagent-Red (ACD) following the manufacturer's instructions. The slides were denatured at 60 °C for 1 h and incubated with pre-warmed (60 °C) SIV gag-pol sense probe (Cat No: 312811, ACD) with an additional incubation for 4 h at 40 °C. The signal was then amplified by Amp1 to Amp6 and identified on tissue sections using a Zeiss digital slide scanner system.

### In vivo CD8+ lymphocyte depletion

To deplete CD8+ lymphocytes (CD8+ T cells and NK cells), rhesus macaques received a single dose of 50 mg/kg rhesus IgG1 recombinant Anti-CD8α monoclonal antibody MT807R1 (Cat No: PR-0817, NIH NHP Reagent Resource) by intravenous route[102]. Absolute CD8+ T-cell counts (Ab clone, SK1 and RPA-T8), and viral loads were measured following MT807R1 infusion.

### ELISA

ELISA plates were coated with SIVmac239 gp140 (Cat No: IT-001-140p, Immune Tech, New York, NY) at 5 μg/mL in PBS at 4 °C overnight. After blocking with 1% BSA in PBS at 37 °C for 1 hour, serially diluted plasma was incubated on plate at 37 °C for 1 hour. 1:1000 diluted Horseradish peroxidase (HRP)-conjugated goat anti-human IgA+IgG+IgM (Cat No: 109-035-064, Jackson ImmunoResearch Laboratories, Inc., West Grove, PA) was added at 37 °C for 1 hour. All volumes were 100 μL/well, except 200 μL/well for blocking. Plates were washed between each step with 0.1% Tween 20 in PBS, developed with 3,3´,5,5´-tetramethylbenzidine (TMB) (Cat No: 860336, Sigma-Aldrich), and read at 450 nm.

### SIV neutralization antibody assay

SIV neutralization was measured using single-round infection of TZM-bl cells with Env pseudoviruses[103]. Briefly, 40 μL of virus was incubated for 30 min at 37 °C with 10 μL of serially diluted plasma in duplicate wells before the addition of TZM-bl cells. To maintain consistent assay conditions, sham medium was used in place of plasma in specified control wells. Plasma dilutions were defined at the point of incubation with virus supernatant. Virus infection levels were determined after 2 days by a luciferase assay (Promega, Madison, WI). Neutralization curves were fitted by the 5-parameter nonlinear regression built in Prism 9.0 (GraphPad Software, La Jolla, CA), and 50% inhibitory dilutions ($ID_{50}$) were defined as the plasma reciprocal dilutions required to inhibit viral infection by 50%.

### Statistical analysis

Statistical analyses were performed by GraphPad Prism 9.0 Software (GraphPad Software, San Diego, CA). Statistical comparison between groups at different timepoints was analyzed using Mann–Whitney test. A nominal $\alpha$ level of 0.05 was used to define statistical significance and the data are presented as mean and SEM. The relationship of

integrated viral reservoir size in tissues with the timing after viral inoculation was analyzed using a nonlinear regression curve fitting model.

**Reporting summary**

Further information on research design is available in the Nature Research Reporting Summary linked to this article.

## Data availability

Source data are provided with this paper in the Source Data file and Supplementary Information.

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

## Acknowledgements

This work was supported by National Institutes of Health grants R01 DE025432, R01 AI147372, R01 HD099857, and the Office of Research Infrastructure Programs (ORIP) grant no. OD011104. We thank Eddie Xu for the editing. Antiviral drugs were kindly provided by Gilead Sciences (TAF, FTC) and ViiV Healthcare (DTG). The Anti-CD8 alpha [MT807R1] antibody used in this study was provided by the NIH Non-human Primate Reagent Resource (P40 OD028116).

## Author contributions

X.Wang, E.V., S. Siddiqui, K.T., H.L., R.B., X.Wu, M.W., W.Z., J.S., L.D.M., K.E. Russell-Lodrigue contributed to data collection. R.P.B., R.S.V. contributed to review & editing. X.W. and H.X. designed the experiments, supervised the projects, interpreted data, and wrote the manuscript.

## Competing interests

The authors declare no competing interests.
