## [Peer review file · Nature Communications]

REVIEWER COMMENTS

Reviewer #1 (Remarks to the Author):

Overall, the manuscript by Wang et al., presents very interesting data demonstrating the efficacy of early ART in neonates to effectively suppress virus (SIV/HIV), even after the removal of ART. The animal studies demonstrate that the initiation of ART 3 days after IV inoculation of SIV, with continuation of ART for 9 months, prevented viral rebound after ART discontinuation in 80% of animals. These observations are significant and are important contributions to the field. However, the conclusion raised and the interpretation of the data, as related to the establishment of reservoirs, are not well-justified.

The study examines the dynamics of early ART on the establishment of SIV reservoirs by quantifying cell-associated RNA and DNA, as well as integrated DNA using an Alu-PCR method. They found that integrated viral DNA did not appear until day 3, although total SIV DNA was found on day 1 in lymph node, colon and PBMC. The study equates efficacy of early ART with the presumption that integrated reservoirs are not established until at least day 3 (lines 90-96, 160, and much of the discussion). There are several problems with this interpretation. While it is possible that unintegrated virus can contribute to the production of plasma virus, the early and rapid kinetics of viral expression in the plasma between days 1 and 3 are very likely to result from integrated provirus. It has not been shown that unintegrated virus would significantly contribute to the early (and rapid) kinetics of viral expression observed following IV inoculation. It is highly plausible that the inability to detect integrated provirus at day 1 or day 3 is due to the sensitivity of the Alu-PCR assay. It is critical to demonstrate the sensitivity of this assay and show that it has the same level of sensitivity as the total DNA to support this claim. The method section does not adequately detail the differences in the integrated and total SIV DNA assays. The authors indicate a sensitivity of the nested, qPCR alu-SIV qPCR assay as 1 copy in 240,000 cells, but they do not compare that with the sensitivity of the total SIV DNA assay nor detail how this was determined/confirmed. The standard used for quantification of integrated DNA is not clear and sufficient details (or a reference) for the sensitivity of the integrated DNA assay are not provided, including the total amount of cell DNA (cell equivalents analyzed).

Despite the limitations regarding their interpretation of integrated provirus, the paper does detail novel and important data regarding the efficacy of early ART in a pediatric population.

Below are a few specific comments that would enhance the manuscript.

Revise all discussion regarding the evidence for lack of reservoir establishment, with the inclusion of assay sensitivity and interpretation of results reported with these limitations.

Figure 1:

The plasma viral load data in Figure 1B is not complete (only shows 6 of 14 animals), and interpretation is limited. It would be helpful to know the plasma viral load of the animals that were sacrificed, and that is not easily discerned from data shown on days 1 and 3.

Plasma VL at day 1 is shown for 6 animals, 3 positive and 3 negative. Which of these are from the animals euthanized and correspond to the data shown from PBMC? What subsequent data is available for the 3 negative? Do the viral loads at day 3 represent the 8 remaining animals? It is difficult to count the number of symbols shown for Day 3; are 8 remaining animals represented in 1B? How do specific plasma VL compare to those in tissues? Is there a threshold of plasma virus levels that correlate with detection of total DNA and/or integrated DNA (suggesting sensitivity of detection methods?) Why do the y axis for the integrated DNA in panels I-M start at 10^{-2} ?

Figure 2:

The legend/text state that 9 control animals were used for comparison, yet only 3 are represented in panels E, F, G. How many are represented in the various time points of panel D, as error bars are only shown for the day 9 time point?

Reviewer #2 (Remarks to the Author):

In this manuscript Wang et al. assessed the impact of very early antiretroviral initiation on the viral reservoir in perinatally SIV-infected rhesus macaque infants. Specifically, the study investigated SIV DNA and RNA levels in RM infants infected i.v. with SIVmac251 within 6 hours after birth and treated or not with ART initiated between day 3 and day 5. The impact of early ART on the viral reservoir was estimated by measuring total or integrated SIV DNA in cells isolated from blood, lymph nodes and gut and/or by analytical treatment interruption. The main result of this study is the long-term control of SIV replication following treatment interruption in 4 out of 5 rhesus macaque infants initiated on ART 3 days post infection and maintained on ART for 9 months. This absence of viral rebound was observed for up to 18 months even following experimental CD8⁺ T cell depletion.

The manuscript needs serious English editing. It also suffers from the choice of graphical representation— at best the graphs are difficult to read but they also can be misleading. The assays used to evaluate reservoir seeding are not optimal and the methods are insufficiently described. The data are not appropriately described and interpreted. The significance of the results is questionable notably because they are not very novel. It is well documented that early ART restricts the latent reservoir

establishment in adults and infants. Delayed viral rebound or sustained virologic control following ART interruption has been reported in children treated early post infection. The Mississippi baby treated 31 hours after birth maintained undetectable viremia for 27 months in absence of ART. Preliminary results from the IMPAACT P1115 study showed that two years after starting ART within 48 hours of birth, two thirds of infants had nondetectable cell-associated HIV DNA and almost 90% tested negative for HIV antibody. ATI will be performed in eligible study participants.

Please see specific comments and detailed review below:

1- To interrogate SIV dynamics or the viral reservoir seeding, the accuracy and sensitivity of the assays used is critical. In this manuscript, some key information regarding the methods is missing such as the limit of detection of each assay and the number of cells assessed (known limiting factor in pediatric studies). The type of cells assessed is also unclear to this reviewer (PBMC and lymphocytes/CD3+ isolated from tissues or tissue mononuclear cells?). If I understand correctly, dilutions of a plasmid in DNA from uninfected cells was used as a standard for the integrated SIV DNA quantification which bypasses the Alu-gag PCR and is thus not appropriate. The manuscript reports unintegrated SIV DNA which has not been measured but extrapolated from the measurement of total and integrated SIV DNA (total SIV DNA in that case is not shown).

2- Untreated SIV infection (Figure 1). The number of animals euthanized or initiated on ART for each time point is not indicated on the study design schematic nor in its legend. The LOD of the assays is missing for total SIV RNA and integrated DNA. It is also unclear if PVL was assessed for all living animals at each time point. Using individual symbols for each animal would greatly improve the figure. I count 9 symbols of the PVL at day 3 but according to the study design, only 8 animals were still alive by then.

3- Early short-term ART (Figure 2): It is unclear what comparisons are the p values representing although the legend states that comparison were made with d3 p.i. Surprisingly, SIV RNA levels appear to be higher in the group of animals receiving ART at day 5 p.i. than in the untreated control group. Were those all assessed at the same time point (day 28 p.i.)? The results indicate that in untreated infection, viremia plateaued at 7 day p.i. but the graph shows a peak at day 9 p.i.

4- Early "long-term" ART and ATI (Figure 3): A control group initiated on ART later and going through ATI is missing. In panel A, it is difficult to distinguish the PVL of each animal at day 3 p.i. One RM out of 5 experienced a viral rebound following ATI. The rebound was detected 2 months post ATI. Was the PVL assessed during these 2 months? The absence of symbol in panel A does not allow the reader to get that information. The antibody used to assess CD8 expression following experimental depletion is not indicated.

Reviewer #3 (Remarks to the Author):

One restatement of the conclusions of this work that could be drawn from this observation is that the authors have not induced HIV remissions, but simply observed successful post-exposure prophylaxis, and that in the setting of new infection of the infant immune system, that potent ART that includes an INSTI within 3 days, but not later, of viral exposure is sufficient to block all persistent infection in 80% of exposed infant animals. There may have been viral integration events that led to productive infection before day 3, but few enough so that no latent, persistent infections were established that could lead to viral rebound upon later ATI.

I view this work as suggestive, but preliminary. A larger cohort should be studied. Longer ATI and more tissue sampling should be pursued. Cell infection transfer experiments, to test if infection can be passed by cells from one animal to another should be done. These points should be discussed by the authors.

However, the findings of this work are potentially important and should be disseminated to stimulate the field.

Current US guidelines (Dec. 2021) state that:

A newborn's ARV regimen should be determined based on maternal and infant factors that influence the risk of perinatal transmission of HIV (AII). The uses of ARV regimens in newborns include the following:

- ARV Prophylaxis: The administration of one or more ARV drugs to a newborn without documented HIV infection to reduce the risk of perinatal acquisition of HIV.
- Presumptive HIV Therapy: The administration of a three-drug ARV regimen to newborns who are at highest risk of perinatal acquisition of HIV. Presumptive HIV therapy is intended to be preliminary treatment for a newborn who is later documented to have HIV, but it also serves as prophylaxis against HIV acquisition for those newborns who are exposed to HIV in utero, during the birthing process, or during breastfeeding and who do not acquire HIV.
- HIV Therapy: The administration of a three-drug ARV regimen at treatment doses (called antiretroviral therapy [ART]) to newborns with documented HIV infection (see Diagnosis of HIV Infection in Infants and Children).

These findings suggest that such presumptive HIV therapy or therapy given for HIV infection documented at birth should include an INSTI if possible. Further support for the findings in this manuscript could lead to the study of ATIs in newborns treated with INSTIs.

Such studies are underway but it is unclear if they are sufficiently enrolled or powered to answer the question anytime soon. Therefore, even these preliminary findings are of interest.

Minor points:

Line 59: Protease inhibitors for “proteinase inhibitors”

Line 59-60: “prevent cells from viral genome integration” is awkward. Change to “prevent viral genome integration in cells” ?

Line 201: typographical error: Integrated

RE: NCOMMS-22-04053, point to point responses to reviewer comments

Reviewer #1

1. Overall, the manuscript by Wang et al., presents very interesting data demonstrating the efficacy of early ART in neonates to effectively suppress virus (SIV/HIV), even after the removal of ART. The animal studies demonstrate that the initiation of ART 3 days after IV inoculation of SIV, with continuation of ART for 9 months, prevented viral rebound after ART discontinuation in 80% of animals. These observations are significant and are important contributions to the field. However, the conclusion raised and the interpretation of the data, as related to the establishment of reservoirs, are not well-justified.

Response: We appreciate the insightful comments that the reviewer provided and thank you for the interest showed in our work. We have accommodated the requests and thoroughly revised the manuscript as suggested.

2. The study examines the dynamics of early ART on the establishment of SIV reservoirs by quantifying cell-associated RNA and DNA, as well as integrated DNA using an Alu-PCR method. They found that integrated viral DNA did not appear until day 3, although total SIV DNA was found on day 1 in lymph node, colon and PBMC. The study equates efficacy of early ART with the presumption that integrated reservoirs are not established until at least day 3 (lines 90-96, 160, and much of the discussion). There are several problems with this interpretation. While it is possible that unintegrated virus can contribute to the production of plasma virus, the early and rapid kinetics of viral expression in the plasma between days 1 and 3 are very likely to result from integrated provirus. It has not been shown that unintegrated virus would significantly contribute to the early (and rapid) kinetics of viral expression observed following IV inoculation. It is highly plausible that the inability to detect integrated provirus at day 1 or day 3 is due to the sensitivity of the Alu-PCR assay. It is critical to demonstrate the sensitivity of this assay and show that it has the same level of sensitivity as the total DNA to support this claim. The method section does not adequately detail the differences in the integrated and total SIV DNA assays. The authors indicate a sensitivity of the nested, qPCR Alu-SIV qPCR assay as 1 copy in 240,000 cells, but they do not compare that with the sensitivity of the total SIV DNA assay nor detail how this was determined or confirmed. The standard used for quantification of integrated DNA is not clear and sufficient details (or a reference) for the sensitivity of the integrated DNA assay are not provided, including the total amount of cell DNA (cell equivalents analyzed).

Response: We appreciate the reviewer's concerns. To address the concerns regarding the early virion production from unintegrated viral DNA and the sensitivity of the qPCR assay, we performed additional *in vitro* experiments, in combination with viral parameters measured by nested (*Alu*) qPCR. Briefly, PBMCs isolated from one-month-old infant macaques (due to limited blood collected from newborns or neonates even at animal necropsy) were first activated for 3 days by CD4+ T

cell activation kit (Miltenyi) in the presence of human IL-2 (10%, Hemagen), and then infected with SIVmac251 (3×10^7 cell/mL with 100 TCID₅₀ SIV) for 2 hours in CO₂ incubator, followed by 3 washes and seeding in 24-well plate at 1×10^6 cell/well/mL, without or with raltegravir (RTG) at final 1 μ M concentration. Cells and cell supernatants were harvested to measure CA SIV RNA/DNA and supernatant viral load. Please see the revised Method section in the revised manuscript.

As shown in **Suppl Fig.1**, cell-associated (CA) integrated SIV DNA was indeed not detected in PBMCs in presence of RTG, yet total CA SIV RNA/ DNA were still detectable as early as 6 hours after SIV infection, suggesting that viral genome integration was fully blocked by RTG treatment while total SIV DNA predominantly represented unintegrated form at such scenario. Please note that supernatant VLs have not shown significant differences at early SIV infection (3 and 7dpi) between RTG treated and untreated samples. This suggests early virus particles in cell supernatants are not likely produced from integrated SIV DNA when PBMCs were treated by RTG. Although integrated proviral DNA is believed a template for stable HIV/SIV replication, however, unintegrated viral DNA *per se* may contribute to the early viral replication and virus production, even when cells are treated by RTG (Kelly, 2008; Chan CN, 2016; Trinite B, 2013), explaining viremia emergence in very early SIV infection *in vivo* prior to viral genome integration. Again, these results suggested that, i) *Alu* qPCR assay here could essentially reflect the status of SIV genome integration; ii) integrase inhibitor treatment could efficiently suppress viral genome integration; iii) viral load in cell supernatants was detectable at early infection in presence of integrase inhibitor (lack of integrated viral DNA), indicating early unintegrated viral DNA is able to rapidly yield infectious virions.

We would prefer to add even more detail in Methods but have space limitations. However, these assays are based on prior studies of integrated SIV DNA measurement which target one conserved region of human and rhesus common *Alu* DNA sequence by outward primer Alu-1 and -2 (Nishimura Y, 2009; Mavigner M, 2016; Whitney JB, 2014 and 2018). Combined unique rhesus *Alu* repetitive sequences (Hamdi H, 1999), we further optimized and improved the *Alu* qPCR to target multiple *Alu* islands, combined Alu-1, -2 with additional rhesus specific Alu -3 and -4 primers, potentially increasing the sensitivity and detection rate of integrated SIV DNA. Human CEMx174 Cell line (#ARP-13239, NIH AIDS Reagent Program), carrying 1 copy of integrated SIV DNA, is used to extract cellular genomic DNA, which is serially diluted to determine copies of proviral DNA in terms of cell numbers, and then quantified by our nested *Alu* qPCR. Unfortunately, there is no rhesus-derived cell line that contains single copy of SIV proviral DNA and specific rhesus *Alu*, thus we cannot test human CEMx174 Cell lines by our new *Alu* primers (Alu 3- and -4). However, our nested qPCR with *Alu* 1-4 primers generates consistently higher levels of proviral DNA in macaque-derived cell samples, compared with of those using human/rhesus *Alu* 1-2 only. Limit of detection (LOD) of proviral SIV DNA is 5.3 per million cells, determined by GenEx software based

Table 1. Nested qPCR reproducibility (%CV; n=6)		
STD Copies	CCR5	LTR U5
10 ⁴	0.2%	2.9%
10 ³	0.3%	2.5%
10 ²	0.4%	2.1%
10 ¹	0.5%	2.4%
10 ⁰	1.2%	3.5%

on standard curve and deviation, please see %CV for individual standard in **Table 1** in this letter. In our assay, lymphocytes (1×10^7 cells per sample in general) are isolated to extract cellular DNA (final volume to 100 μ L) and 2.5 μ L cellular DNA sample (2.5×10^5 cell equivalents) is added to the Master Mix (final 25 μ L). The data showed that 1 copy SIV genomic DNA contained cellular DNA equivalent to cell numbers,

ranging from 1×10^5 to 2.4×10^5 cell equivalents, could be consistently detected (stated probability) as detection quantification (LOQ, 1 copy in 2.4×10^5 cell equivalents, %CV=4.2%). In brief, first-step routine PCR (pre-amplification) is performed to amplify SIV LTR U5 (for total SIV DNA and standard) or SIV U5/Alu 1-4 fragments (for proviral DNA) for 20 cycles, followed by second-step qPCR using same U5 primer/probe set for 40 cycles with 1/10 volume

of 1st amplicon. Internal standard curve was used to calculate the initial copies of both total SIV DNA and proviral DNA (an example of STD curve attached here, very stable from batch to batch).

Using this assay, we have measured hundreds of samples from SIV-infected animal cohorts (Ziani W, 2021^{a,b,c}; Wang X, 2021). We performed nested qPCR in parallel to quantify both total viral DNA and proviral DNA in each sample (yielding comparable data), along with standard. Thus, nested qPCR assay can generate sensitive and comparable data of both total SIV DNA and proviral DNA. Also, the results shown in Supple Fig. 1 suggest that *Alu* qPCR is able to essentially reflect the sensitivity and status of SIV genome integration in RTG- or untreated-PBMCs, thus inferring lymphocyte samples isolated from tissues. Basically, nested qPCR (two-rounds PCR amplification) can increase sensitivity and specificity of target gene, however, we admit that the possibility that proviral DNA may not be fully detected due to viral reservoir distribution in various tissues in the body and likely lack of chromosomal *Alu* islands situated near the integrated SIV genome, we discussed this in the revised DISCUSSION. *Alu* islands are widely dispersed within the chromosomal DNA (accounting for 6-13% of genomic DNA), which are likely lacked or far from SIV LTR region, albeit very few such case existed. qPCR targeting SIV *gag* is used to determine plasma viral load with a limit of detection of 83 copies per mL (core service). Together, early treatment, building on viral genome integration assessed by our assay, achieved sustained virologic remission in the neonates infected with SIV, which were largely consistent with the onset of proviral reservoirs in typical systemic and lymphoid tissues of very early SIV infection. We realize that more animal cohorts/sampling are needed, in combination with intrauterine/postnatal infection, one or more ARV regimens as well as optimal period of treatment and so on. We have carefully considered the interpretations of our results and added more information and detail in Method section, please see in our revised manuscript.

3. Despite the limitations regarding their interpretation of integrated provirus, the paper does detail novel and important data regarding the efficacy of early ART in a pediatric population.

Response: We thank reviewer's positive comments, we have carefully read through the text and made appropriate interpretations and caveats to our results.

4. Revise all discussion regarding the evidence for lack of reservoir establishment, with the inclusion of assay sensitivity and interpretation of results reported with these limitations.

Response: We thank reviewer's suggestion, we discussed proviral reservoir establishment in neonates, especially assay sensitivity and interpretation of proviral reservoir measurement *in vitro* and *in vivo* studies, please see DISCUSSION in our revised manuscript.

5. *Figure 1: The plasma viral load data in Figure 1B is not complete (only shows 6 of 14 animals), and interpretation is limited. It would be helpful to know the plasma viral load of the animals that were sacrificed, and that is not easily discerned from data shown on days 1 and 3.*

Response: We apologize for not making it clearer. Whole blood request and neonatal anesthesia is strictly controlled in terms of IACUC (12mL/Kg/Month in total) and limited volume of experimental blood available because of newborns weight (~ 400g, only 1.2cc EDTA blood available in general), physiologic weight loss after birth as well as mandatory bleeding for CBC. It is not feasible to consistently collect 1.2cc whole blood from newborn infants and neonates in the first few days of birth and larger volumes are limited to collection at necropsy. Therefore, whole blood is collected by staggered request from individual newborn/neonates. We added the grouped animals, the day(s) post SIV inoculation, plasma viral load from blood sampling as well as text in revised manuscript as **Suppl. Table 1**. Hope it will be clearer.

6. *Plasma VL at day 1 is shown for 6 animals, 3 positive and 3 negative. Which of these are from the animals euthanized and correspond to the data shown from PBMC? What subsequent data is available for the 3 negative? Do the viral loads at day 3 represent the 8 remaining animals? It is difficult to count the number of symbols shown for Day 3; are 8 remaining animals represented in 1B? How do specific plasma VL compare to those in tissues? Is there a threshold of plasma virus levels that correlate with detection of total DNA and/or integrated DNA (suggesting sensitivity of detection methods?) Why do the y axis for the integrated DNA in panels I-M start at 10⁻²?*

Response: We apologize for the confusion. Here we improved the figures to specifically track cell-associated (CA) SIV RNA/DNA in PBMC, LN and colon, and corresponding plasma viral load (PVL) in individual animals at early SIV infection by colored symbols. As we mentioned in Question #5 and shown in **Suppl. Table 1**, plasma viral load measured at 1dpi (n=6) also include staggered blood samples collected from 3 animals that were euthanized at 28dpi (n=3; PVL, 1 negative and 2 positive at 1dpi), the remaining three animals euthanized at 1dpi are highlighted by colored symbols now to accommodate plasma viral load (2 negative and 1 positive), CA SIV RNA/DNA in PBMC, LN and colon in individual animals. Likewise, viral parameters from neonates euthanized at 2dpi (n=3), 3dpi (5 euthanized neonatal animals plus additional 3 staggered blood samples from other 28-day infected animals for PVL measurement, see Suppl. Table 1), 5dpi (n=2) and 7dpi (n=1) are highlighted by colored symbols for individual animals post SIV infection in revised manuscript (Figs. 1B-1H). As for sensitivity and LOD of qPCR detection, please see Answer # 2. Given y-axis represents power of 10, value below the LOD is considered to be undetectable, we now revised the figures. We added necessary description in Results, and hope it is more understandable now.

7. *Figure 2: The legend/text state that 9 control animals were used for comparison, yet only 3 are represented in panels E,*

F, G. How many are represented in the various time points of panel D, as error bars are only shown for the day 9 time point?

Response: We are sorry for our carelessness. We longitudinally tracked plasma viral load in plasma including staggered blood samples (Figs. 2A-2C), CA SIV RNA and paralleled CA SIV DNA/proviral DNA in fresh tissues of 3 groups on cART initiated at 3-, 4- or 5dpi, and one group without treatment control (n=3), all these animals are euthanized at 28dpi. Additional data (not samples) from our previous study with age-matched SIV-infected infant macaques (n=9) are used to compare routine plasma viral load and the percentages of peripheral CD4+ T cells (Fig. 2D and Figs. 3A-3B). We went through the data and revised our text and Figure Legend. As for Fig.2D, which is summarized from Figs. 2A-2C with statistical analysis for each group, the error bars cannot be displayed because of low S.E.M in Y-axis at timepoints.

Reviewer #2

In this manuscript Wang et al. assessed the impact of very early antiretroviral initiation on the viral reservoir in perinatally SIV-infected rhesus macaque infants. Specifically, the study investigated SIV DNA and RNA levels in RM infants infected i.v. with SIVmac251 within 6 hours after birth and treated or not with ART initiated between day 3 and day 5. The impact of early ART on the viral reservoir was estimated by measuring total or integrated SIV DNA in cells isolated from blood, lymph nodes and gut and/or by analytical treatment interruption. The main result of this study is the long-term control of SIV replication following treatment interruption in 4 out of 5 rhesus macaque infants initiated on ART 3 days post infection and maintained on ART for 9 months. This absence of viral rebound was observed for up to 18 months even following experimental CD8+ T cell depletion.

8. The manuscript needs serious English editing. It also suffers from the choice of graphical representation— at best the graphs are difficult to read but they also can be misleading. The assays used to evaluate reservoir seeding are not optimal and the methods are insufficiently described. The data are not appropriately described and interpreted. The significance of the results is questionable notably because they are not very novel. It is well documented that early ART restricts the latent reservoir establishment in adults and infants. Delayed viral rebound or sustained virologic control following ART interruption has been reported in children treated early post infection. The Mississippi baby treated 31 hours after birth maintained undetectable viremia for 27 months in absence of ART. Preliminary results from the IMPAACT P1115 study showed that two years after starting ART within 48 hours of birth, two thirds of infants had nondetectable cell-associated HIV DNA and almost 90% tested negative for HIV antibody. ATI will be performed in eligible study participants.

Response: We appreciate the reviewer's comments. We have carefully looked over the manuscript and improved the typos, grammar issues and interpretation. We apologize for not making the graphs clearer, especially Fig.1, and have thoroughly revised the figures for plasma viral load and counterpart CA SIV RNA/DNA in individual animals at very early SIV infection as described above. Now neonates used are summarized in Suppl. Table 1. We also added more detailed description regarding our qPCR assays in Methods, including how we quantified the viral parameters using our optimized nested (Alu) PCR based on other published reports (Nishimura Y, 2009; Mavigner M, 2016; Whitney JB, 2014 and 2018), and also please see Answer #2 in this letter. Specifically, we added the supplementary data (**Suppl Fig. 1**) to directly address specificity

and sensitivity of the assay especially for integrated viral DNA measurement, with or without raltegravir (RTG) treatment. As shown in **Suppl Fig.1**, CA integrated viral DNA was consistently not detected in PBMCs in presence of RTG (viral genome integration blockade), while CA SIV RNA/DNA (unintegrated, due to undetectable integrated vDNA) as well as supernatant viral load were still detectable in the early SIV infection. These data strongly suggest that our assay can essentially reflect the status of SIV genome integration, by comparing the results in the presence or absence of RTG treatment, and thus inferring this application to measure the lymphocyte samples isolated from tissues *ex vivo*.

In this study, we utilized the neonatal NHP model intravenously infected with SIV, facilitating to precisely time the very early days of infection (compared with uncertain dpi by repeated SIV/SHIV challenges to the infant macaques by oral route or HIV+ children in the clinic) and collected complete tissue sets at necropsy for examination of viral reservoirs. As shown in **Fig.2** in this letter, early cART initiation at 3dpi in adult animals showed 100% viral rebound within three weeks once treatment is discontinued (*Whitney, 2018*). In comparison, early treatment regimen initiated at same timepoint achieved sustained virologic remission in 80% infant macaques, suggesting the *features and outcomes in neonates/infants to the early SIV/HIV infection and early treatment are indeed distinct from those of adult subjects, which is of great significance to understand early events and providing a critical treatment window for pediatric functional cure or even absolute cure.*

It is true that early ART in children, treatment ranging from immediate (*Butler, 2015*), within hours after delivery (*Giacomet, 2014*), or months/years-old (*Martinez-Bonet, 2015; Frange, 2016; Violari, 2019*), is documented well, including reduction of the reservoir size and skewing of viral rebound in the clinic setting. However, *viral rebound or rapid increase of HIV RNA/DNA in these early treated pediatric cases are detected once treatment is stopped in these clinical trials.* The Mississippi baby case treated 30 hours after birth ultimately showed viral rebound even after 27-months ATI, *in utero* HIV infection during pregnancy likely occurred in this case (*Faye A, 2020*), and conceivably, viral integration occurred in perinatal period. We do not know either status of intrauterine infection even starting ART within 48 hours of birth, and based on what happened if treatment is allowed to withdraw in IMPAACT P1115 study, the outcomes presumably depend on status of viral genome integration.

In this study, we evaluated the dynamics and tissue distribution of viral reservoirs and plasma viral load in several newborn macaques at very early timed-stages of SIV infection, which provided critical clues for early treatment interventions, as indicated by outcomes of viral remission achieved by early treatment for 9 months and further confirmed by subsequent CD8+ cell depletion *in vivo*. We consider this a *de facto* success to achieve viral remission in 4/5 infant macaques by early treatment (all 5 infants infected with identical SIV inoculum at birth and treated at 3dpi for 9 months). These 4 cured infants are still alive with ART-free viral remission, and currently resisting repeated SIV challenges by the rectal route. Here we

precisely explored the neonatal distinct patterns, early viral reservoir seeding and optimal early treatment (timing and regimen) utilizing the neonatal macaque model of HIV, which permits thorough examinations of blood and tissue reservoirs that cannot be achieved studying human infants. Even if this data shows only a few unique findings we believe this data is directly translational to the clinic and may provide knowledge that may save a few infants from lifelong HIV infection.

9. To interrogate SIV dynamics or the viral reservoir seeding, the accuracy and sensitivity of the assays used is critical. In this manuscript, some key information regarding the methods is missing such as the limit of detection of each assay and the number of cells assessed (known limiting factor in pediatric studies). The type of cells assessed is also unclear to this reviewer (PBMC and lymphocytes/CD3+ isolated from tissues or tissue mononuclear cells?). If I understand correctly, dilutions of a plasmid in DNA from uninfected cells was used as a standard for the integrated SIV DNA quantification which bypasses the Alu-gag PCR and is thus not appropriate. The manuscript reports unintegrated SIV DNA which has not been measured but extrapolated from the measurement of total and integrated SIV DNA (total SIV DNA in that case is not shown).

Response: We appreciate the reviewer's concerns. We absolutely agree with the opinion that the viral reservoir assays are highly challenging for the current studies and researchers in general, as indicated by continuous efforts and attempts made for development of scalable assays, including RT qPCR, *Alu* qPCR, QVOA, TILDA, IPDA and FLIPS, *etc.* As mentioned in DISCUSSION, we acknowledge that there is no *de facto* gold standard thus far to reflect absolute size of viral reservoirs especially viral DNA (e.g., unintegrated, or integrated vDNA, circular vDNA, defective or intact viral genome, functional or dysfunctional, Indel/polymorphism at different levels, inserted regions with gene desert, etc), with obvious both advantages and weaknesses with each approach. To our knowledge, it is a challenge to measure integrated HIV/SIV DNA efficiently, comprehensively, and accurately in the cells, tissues, and organs throughout the body, while most have only assessed peripheral blood mononuclear cells, and/or even sorted CD4+ T cells thus far, which obviously misses important residual reservoirs in tissues.

Considering sensitivity, rapidity, and feasibility of qPCR, we spent several years to develop and optimize nested *Alu* qPCR for quantification of the CA integrated SIV DNA, based on previous published reports using conserved human/rhesus *Alu*-1 and -2 (*please see refs by Nishimura Y, 2009; Mavigner M, 2016; Whitney JB, 2014 and 2018, etc*). To strengthen proviral SIV DNA assay, rhesus specific repetitive *Alu* DNA, which are widely dispersed within the chromosomal DNA in rhesus macaques, is targeted by additional *Alu*-3 and -4 primers. Our optimized *Alu* qPCR thus includes *Alu* 1-4 outward primers to target multiple *Alu* islands in rhesus species. As for the accuracy and sensitivity of the assays concerned, please see Answer#2.

It is true that we use plasmids as standard containing SIV gene as standard with known copies, which is indispensable to perform nested qPCR (unlike the RNA quantification by Droplet digital PCR, STD not required). Human CEMx174 Cell line (#ARP-13239, NIH AIDS Reagent Program), carrying 1 copy of integrated SIV DNA, is generally used to quantify proviral DNA by nested *Alu* qPCR with conserved human/rhesus *Alu*-1 and -2 outward primers. Additional *Alu* primers (*Alu*-3 and -4), targeting rhesus specific *Alu* sequences, are not applicable for human CEMx174 Cell line-associated

standard. Outward Alu 1-4 primers used in qPCR indeed increase the sensitivity and detection rate of proviral SIV DNA in rhesus samples. To yield comparable data in the quantification of total SIV DNA and proviral DNA, nested qPCR is performed in parallel to quantify both total SIV DNA and proviral DNA for each sample. Type of cells are lymphocytes that are freshly isolated from systemic and lymphoid tissues at biopsies or necropsies, 10^7 lymphocytes per sample are generally collected for nested qPCR assay. Briefly, we performed nested qPCR to quantify total viral DNA, proviral DNA, along with standard. First-step routine PCR is used to pre-amplify total SIV DNA and standard (by LTR U5 primer pair) or integrated SIV DNA (by U5/Alu 1-4 primer pairs) for 20 cycles (optimized cycle for the 1st PCR in our assay). Second qPCR is further performed with 1/10 volume of 1st amplicon targeting SIV LTR U5 with same primer/probe set for 40 cycles. Internal standard curve is used to calculate the initial copies of both total and integrated SIV DNA. Please note that it is essentially difficult to quantify gene fragments of SIV/Alu by qPCR in the first round PCR due to unpredictable size of gene fragments generated (tens of bp to several kb) and thereby amplification efficiency is variable from sample to sample. Thus, it is feasible and reasonable to use plasmid standard in the quantification of both CA total SIV DNA and proviral DNA at 2nd qPCR *using same primer/probe set, irrelevant to Alu repetitive sequences*. We agree with reviewer's comments about unintegrated SIV DNA, now we changed wording to indicate total SIV DNA in the revised figures.

Although our assay is far more sensitive to measure proviral DNA, nested qPCR increases sensitivity and specificity of gene targets, as also indicated by Ct (*e.g.*, cycle threshold of 10copies, from ~36 at 1st to ~16 at 2nd PCR). However, status of viral genome integration by our assay essentially matches the actual scenario: undetectable proviral DNA in PBMCs infected with SIV in presence of RTG, and achievable ART-free viral remission in infants when early cART is initiated prior to proviral reservoir seeding. We have thoroughly modified our text as suggested in the revised manuscript.

10. Untreated SIV infection (Figure 1). The number of animals euthanized or initiated on ART for each time point is not indicated on the study design schematic nor in its legend. The LOD of the assays is missing for total SIV RNA and integrated DNA. It is also unclear if PVL was assessed for all living animals at each time point. Using individual symbols for each animal would greatly improve the figure. I count 9 symbols of the PVL at day 3 but according to the study design, only 8 animals were still alive by then.

Response: We appreciate the reviewer's suggestion and have improved the figure as suggested to track individual animals. PVL is measured in plasma from euthanized animals or staggered blood samples. The LOD of assay is added in the revised manuscript. Also please see the Answer #5 and #6 above in this letter including animals used and corresponding PVL at different timepoints (**Suppl. Table 1**).

11. Early short-term ART (Figure 2): It is unclear what comparisons are the p values representing although the legend states that comparison were made with d3 p.i. Surprisingly, SIV RNA levels appear to be higher in the group of animals receiving ART at day 5 p.i. than in the untreated control group. Were those all assessed at the same time point (day 28 p.i.)? The results indicate that in untreated infection, viremia plateaued at 7 day p.i. but the graph shows a peak at day 9 p.i.

Response: We thank the reviewer for these comments and for not making comparisons clearer among the groups. We revised figures and legend in the manuscript. We did see levels of CA SIV RNA are higher at 28dpi by late treatment. Conceivably, cART (FTC/TFV/DTG) suppresses reverse transcription of viral RNA and viral integration, late treatment may result in accumulation of intracellular SIV RNA which are transcribed from both unintegrated and proviral DNA (*e.g.*, before treatment at 5dpi), while CA SIV RNA in untreated control may be relatively exhausted by virus packaging/release, contributing to viral peak and thereafter. All samples examined are freshly collected from euthanized animal cohorts at 28dpi (Figs. 2E-2F). PVL data in age-matched SIV-infected infant controls come from our previous studies. PVL in SIV-infected neonatal macaques is generally monitored at 7, 14 and 21dpi, viremia actually reaches the peak at 7-9 dpi, we have corrected the description.

12. Early “long-term” ART and ATI (Figure 3): A control group initiated on ART later and going through ATI is missing. In panel A, it is difficult to distinguish the PVL of each animal at day 3 p.i. One RM out of 5 experienced a viral rebound following ATI. The rebound was detected 2 months post ATI. Was the PVL assessed during these 2 months? The absence of symbol in panel A does not allow the reader to get that information. The antibody used to assess CD8 expression following experimental depletion is not indicated.

Response: We thank the reviewer for these comments and suggestions. Neonates and infants infected with HIV/SIV show higher viremia and rapid disease progression compared to adults, most of infant macaques intravenously infected with SIV develop to the final stage of AIDS after 4-5 months of infection (Wang X, 2010). These infants are sacrificed because of AIDS-like symptoms, so the PVL data are shown only in this period. In panel A, PVL levels are very closed in neonatal macaques at 3dpi (Please see PVL of other neonates at this timepoint in **Suppl Table 1**), one RM (NG24, red line) with viral rebound after ATI does not show significantly different levels of plasma viral load compared to remaining 4 animals at this timepoint. Now we showed NG24 PVL line in bold and individual symbols for 5 animals at designated timepoints. All 5 infant animals are weekly monitored by PVL to see if there is viral rebound after ATI. Anti-CD8 Ab clone used to assess CD8+ cells is also added in the revised manuscript.

Reviewer #3

13. One restatement of the conclusions of this work that could be drawn from this observation is that the authors have not induced HIV remissions, but simply observed successful post-exposure prophylaxis, and that in the setting of new infection of the infant immune system, that potent ART that includes an INSTI within 3 days, but not later, of viral exposure is sufficient to block all persistent infection in 80% of exposed infant animals. There may have been viral integration events that led to productive infection before day 3, but few enough so that no latent, persistent infections were established that could lead to viral rebound upon later ATI.

I view this work as suggestive, but preliminary. A larger cohort should be studied. Longer ATI and more tissue sampling should be pursued. Cell infection transfer experiments, to test if infection can be passed by cells from one animal to another should be done. These points should be discussed by the authors.

Response: We thank the reviewer for these thoughtful comments and suggestions. We have since performed experiments

with one-month-old infant PBMC infected with SIV *in vitro*, in presence or absence of integrase inhibitor (raltegravir/RTG), followed by CA SIV RNA/DNA and viral load measurement, which are expected to address the efficiency of RTG on the blockade of viral genome integration, ability of early unintegrated SIV DNA in the virion production as well as sensitivity of qPCR assay, please see the **Suppl. Fig.1** in the revised manuscript. Besides typical systemic and mucosal lymphoid tissues presented, other tissues, including spleen, thymus, jejunum, colon and liver, were actually examined in our study, proviral DNA was not detected in these tissues of very early SIV-infected neonates. We agree with reviewer's constructive advice that more studies are needed, including animal numbers, comprehensive tissue sampling, optimal period of early treatment, prolonged ATI, cell transfer *in vivo* to repeated challenge, *etc*, please see our discussion in the revised manuscript. After CD8+ cell depletion *in vivo*, the 4 infant animals with ART-free viral remission are still alive and healthy, further study with repeated SIV challenge by rectal route is underway. Finally, even if this is only "post exposure prophylaxis" it is clearly different from adults and remains translationally relevant to clinical pediatric HIV care, thus we consider these findings highly significant.

14. However, the findings of this work are potentially important and should be disseminated to stimulate the field.

Current US guidelines (Dec. 2021) state that:

A newborn's ARV regimen should be determined based on maternal and infant factors that influence the risk of perinatal transmission of HIV (AII). The uses of ARV regimens in newborns include the following:

- ARV Prophylaxis: The administration of one or more ARV drugs to a newborn without documented HIV infection to reduce the risk of perinatal acquisition of HIV.*
- Presumptive HIV Therapy: The administration of a three-drug ARV regimen to newborns who are at highest risk of perinatal acquisition of HIV. Presumptive HIV therapy is intended to be preliminary treatment for a newborn who is later documented to have HIV, but it also serves as prophylaxis against HIV acquisition for those newborns who are exposed to HIV in utero, during the birthing process, or during breastfeeding and who do not acquire HIV.*
- HIV Therapy: The administration of a three-drug ARV regimen at treatment doses (called antiretroviral therapy [ART]) to newborns with documented HIV infection (see Diagnosis of HIV Infection in Infants and Children).*

These findings suggest that such presumptive HIV therapy or therapy given for HIV infection documented at birth should include an INSTI if possible. Further support for the findings in this manuscript could lead to the study of ATIs in newborns treated with INSTIs. Such studies are underway but it is unclear if they are sufficiently enrolled or powered to answer the question anytime soon. Therefore, even these preliminary findings are of interest.

Response: We really appreciate the reviewer's encouraging comments for our attempts. Current US guidelines are informative for the preclinical studies utilizing NHP model of HIV. Given our finding that status of viral genome integration at or near the ART initiation may determine the clinical outcomes, we are thus requesting more seasonal newborns, infant macaques of one-month-old and pregnant animals. All these animals will be utilized to investigate the ARV regimens (one or more drugs), fetal and maternal factors, intrauterine infection, INSTI-based prior or after viral integration, optimal timepoint and period of early treatment and outcomes (immune development and viral rebound) *etc*.

Minor points:

15. Line 59: *Protease inhibitors* for “*proteinase inhibitors*”

Response: Corrected.

16. Line 59-60: “*prevent cells from viral genome integration*” is awkward. Change to “*prevent viral genome integration in cells*” ?

Response: Thank you for this helpful suggestion. It has been corrected.

17. Line 201: *typographical error: Integrated*

Response: Corrected.

REVIEWER COMMENTS

Reviewer #1 (Remarks to the Author):

The additions and edits regarding the sensitivity of the SIV DNA and RNA quantification assays have addressed several of the concerns raised in the first review.

There have been some qualifications added to the discussion stating the limitations of the Alu qPCR, however the authors have not demonstrated the sensitivity of the Alu qPCR in "real" samples. Admittedly, this is very difficult to demonstrate, but the assay sensitivity observed with spiked samples does not demonstrate sensitivity in samples with provirus randomly integrated into different cells. The authors also provided in vitro experiments (S1) to demonstrate production of virions, despite absence of integrated virus (with INT administration). These results do not take into account the initial input virus that would still be present in the media.

Collectively, the data does suggest that there are few proviral copies in the early days of infection, but this is to be expected. Without a clear limit of detection for the Alu PCR, it is not possible to make the claim that virions are a result unintegrated viral DNA. Indeed, the early dynamics will be difficult to determine in vivo, as only small samples of PBMC can be analyzed.

Other concerns raised in the initial review have been satisfactorily addressed with the edits.

Reviewer #2 (Remarks to the Author):

The authors have addressed most reviewer's comments and edited manuscript and figures.

ART initiated 3dpi in infant RMs infected with SIV at birth prevented viral rebound following ATI in 4/5 animals. This result is of interest, but interpretations and conclusions must be more cautious. It seems premature to claim an absence of viral integration for 3dpi that would be specific to infants as opposed to adults, notably because the sensitivity of the integrated assay used is not clear, or specific to the ART regimen used as there is no group treated without DTG. And finally, as noted by Reviewer #3, as ART

was initiated within 72h of infection the outcome might be closer to a successful PEP than a cure or remission and this should be discussed.

Please see additional comments below:

Regarding the novel ex vivo experiment presented in supplementing Fig 1, not detecting integration before 5dpi in pre-stimulated PBMC infected in vitro in absence of RTG treatment is surprising and suggest that the integrated assay might be less sensitive than the total DNA one. The response to Reviewer 1 does not provide the sensitivity of the integrated assay nor a comparison of the LOD of this assay with that of the total DNA assay. Similarly, in the in vivo study, it cannot be excluded that integration is not detected before 3dpi in tissues and 5 dpi in PBMC because of a lack of sensitivity of the Alu gag PCR assay. Because the absence of detection of integrated DNA does not necessarily means that there is no integration, the authors should dampen their interpretations and conclusions. This won't take away from the main finding of this study that shows long-term control of SIV replication following treatment interruption in 4 out of 5 rhesus macaque infants initiated on ART 3 days post infection and maintained on ART for 9 months.

I believe there is an error in supplemental table 1 as according to Fig1A no animals were sacrificed 4dpi (I think the animals sampled at 4dpi were sacrificed at 28dpi). This should be corrected. Further, the sentence "The results showed that plasma viral load was detectable in ~50% neonates at 1dpi and subsequent time points thereafter" should be reformulated. After 1dpi all tested samples were positive (not 50%). In the results section, only the first group of animals is introduced "Newborn macaques were intravenously inoculated with identical doses of SIVmac251 within 6 hours after birth, and then euthanized at day 1, 2, 3, 5 and 7 post SIV inoculation for complete tissue collections (Fig. 1A)" before presenting the PVL results of figure 1B that includes animals from the other group of animals (necropsy at 28 dpi after ART). This is confusing. The authors should make clear that PVL were obtained from both groups in absence of ART. The new color coding of the symbols in Figure 1 is not explained in the legend or on the figure. I assume that all animals sacrificed at day 28 have a black symbol and the other ones got a color. I am not sure why the same color was used on different shapes for different animals. This could be made much easier for the readers. At least a key animal ID (as presented in supplemental Fig1)/symbol should be added to this figure.

Comment 8- It has to be noted that while all adult macaques experienced viral rebound following ATI in the Whitney paper, some adult macaques maintained undetectable PVL in the Okoye paper (Nat Med, 2019). While it is clear that the reservoir is seeded early, some differences are seen between models and while the pediatric model might be different, it seems premature to draw definitive conclusions.

Comment 9- Cell type (how they have been isolated), cell number per PCR ("generally 10M") and integration assay sensitivity are still obscure.

Reviewer #3 (Remarks to the Author):

I appreciate the authors efforts to revise their work in response to the comments of my review, and those of other reviewers

RE: NCOMMS-22-04053A, point to point responses to reviewer comments

Reviewer #1

The additions and edits regarding the sensitivity of the SIV DNA and RNA quantification assays have addressed several of the concerns raised in the first review. There have been some qualifications added to the discussion stating the limitations of the Alu qPCR, however the authors have not demonstrated the sensitivity of the Alu qPCR in "real" samples. Admittedly, this is very difficult to demonstrate, but the assay sensitivity observed with spiked samples does not demonstrate sensitivity in samples with provirus randomly integrated into different cells. The authors also provided in vitro experiments (S1) to demonstrate production of virions, despite absence of integrated virus (with INT administration). These results do not take into account the initial input virus that would still be present in the media.

Response: We appreciate reviewer's comments and admit that the Alu qPCR assay may not accurately reflect the *bona fide* seeding and size of viral reservoirs, and that the data based on this assay are not adequate to fully assess levels of potential provirus, so we cautiously interpreted the results. We also realize that protective mechanisms cannot be explained based on these assays, and more animal studies may provide more insight into the relationship of early provirus seeding and cure strategies. To this end, we recently requested newborn macaques for the comparison in the outcomes of treatment regimens with one or more antiretroviral drugs. Although these studies are underway, our primary study showed that plasma viral load was maintained high levels in infants on single DTG initiated at 2dpi (2dpi: $1.8 \times 10^4 \pm 8128$; 7dpi: $7.4 \times 10^4 \pm 25172$; 14dpi: $5.8 \times 10^6 \pm 3748497$, n=3), at least partly demonstrating the contribution of linear viral DNA in virus production under integrase inhibitor treatment. We expect that our further studies and findings could answer these key questions.

Collectively, the data does suggest that there are few proviral copies in the early days of infection, but this is to be expected. Without a clear limit of detection for the Alu PCR, it is not possible to make the claim that virions are a result unintegrated viral DNA. Indeed, the early dynamics will be difficult to determine in vivo, as only small samples of PBMC can be analyzed.

Response: We agree with the reviewer's comments, we sincerely revised description in the manuscript.

Other concerns raised in the initial review have been satisfactorily addressed with the edits.

Response: We appreciate reviewer for the time and comments, and the opportunity to improve our manuscript.

Reviewer #2

The authors have addressed most reviewer's comments and edited manuscript and figures.

ART initiated 3dpi in infant RMs infected with SIV at birth prevented viral rebound following ATI in 4/5 animals. This result is of interest, but interpretations and conclusions must be more cautious. It seems premature to claim an absence of viral integration for 3dpi that would be specific to infants as opposed to adults, notably because the sensitivity of the integrated assay used is not clear, or specific to the ART regimen used as there is no group treated without DTG. And finally, as noted by Reviewer #3, as ART was initiated within 72h of infection the outcome might be closer to a successful PEP than a cure or remission and this should be discussed.

Response: We appreciate reviewer's comments. As mentioned above, we cautiously interpretate the Results and counterpart conclusion, and discuss the aspects of PEP, remission, or cure, please see in our revisions. We also absolutely agree with the suggestions, further *in vivo* studies are needed to fully address the proviruses seeding, early treatment regimens and outcomes in a cure or remission for pediatric HIV infection. Therefore, we recently requested newborn macaques for the different animal groups, treated by one or more antiretroviral drugs administered at the specific early timepoint. We expect that the findings could answer these questions in near future.

Regarding the novel ex vivo experiment presented in supplementing Fig 1, not detecting integration before 5dpi in pre-stimulated PBMC infected in vitro in absence of RTG treatment is surprising and suggest that the integrated assay might be less sensitive than the total DNA one. The response to Reviewer 1 does not provide the sensitivity of the integrated assay nor a comparison of the LOD of this assay with that of the total DNA assay. Similarly, in the in vivo study, it cannot be excluded that integration is not detected before 3dpi in tissues and 5 dpi in PBMC because of a lack of sensitivity of the Alu gag PCR assay. Because the absence of detection of integrated DNA does not necessarily means that there is no integration, the authors should dampen their interpretations and conclusions. This won't take away from the main finding of this study that shows long-term control of SIV replication following treatment interruption in 4 out of 5 rhesus macaque infants initiated on ART 3 days post infection and maintained on ART for 9 months.

Response: We thank the reviewer's comments. Again, we admit that assay *per se*, due to many limitations, is difficult to exclude the possibility of potential proviral DNA existence in samples, so we carefully interpret the results and corresponding conclusion, and focus more on the main findings that 4/5 infant macaques, by early ART at 3dpi up to 9 months, do not show viral rebound after removal of treatment.

I believe there is an error in supplemental table 1 as according to Fig1A no animals were sacrificed 4dpi (I think the animals sampled at 4dpi were sacrificed at 28dpi). This should be corrected. Further, the sentence "The results showed that plasma viral load was detectable in ~50% neonates at 1dpi and subsequent time points thereafter" should be reformulated. After 1dpi all tested samples were positive (not 50%). In the results section, only the first group of animals is introduced "Newborn macaques were intravenously inoculated with identical doses of SIVmac251 within 6 hours after birth, and then euthanized at day 1, 2, 3, 5 and 7 post SIV inoculation for complete tissue collections (Fig. 1A)" before presenting the PVL results of figure 1B that includes animals from the other group of animals (necropsy at 28 dpi after ART). This is confusing. The authors should make clear that PVL were obtained from both groups in absence of ART. The new color coding of the symbols in Figure 1 is not explained in the legend or on the figure. I assume that all animals sacrificed at day 28 have a black symbol and the other ones got a color. I am not sure why the same color was used on different shapes for different animals. This could be made much easier for the readers. At least a key animal ID (as presented in supplemental Fig1)/symbol should be added to this figure.

Response: We thank reviewer to point out and feel sorry to make confused. We corrected the typo in the supplemental table 1 and other descriptions as suggested. To track the viral parameters in individual animal, colored symbols with different shapes are coded. In Figure 1A, different symbol shapes represented plasma viral load of individual animals which were euthanized at day 1, 2, 3, 5 or 7 post SIV infection (colored solid symbols), or from staggered blood samples in infant animals that were sacrificed at 28dpi without treatment (opened black symbols). We added necessary notes and information in the figure legend.

Comment 8- It has to be noted that while all adult macaques experienced viral rebound following ATI in the Whitney paper, some adult macaques maintained undetectable PVL in the Okoye paper (Nat Med, 2019). While it is clear that the reservoir is seeded early, some differences are seen between models and while the pediatric model might be different, it seems premature to draw definitive conclusions.

Response: We thank reviewer's concerns. We also noticed the obviously discrepant findings regarding early treatment and viral rebound off-ART in adult animals, as shown that viral rebound is not observed in 4/16 (ART initiation at 4-5dpi) or 3/17 animals (ART initiation at 6dpi) (Okoye, 2019), essentially contradictory to the reports (100% viral rebound in animals on ART initiation at 3dpi, Whitney, 2018) and clinical setting. It is interesting that total viral DNA is nearly undetectable in both PBMC and lymph node-derived lymphocytes in all SIV-infected adult on early ART throughout the studies (Okoye, 2019), yet only small animal populations ultimately do not show viral rebound after ATI. It is also curious to know the status of proviral DNA in the animal cohorts, unfortunately, proviruses have not been examined. The possible reason may lie in vaccination relevance and sensitivity of assays, despite the reason is still intrigued. We discussed these concerns in revised manuscript.

Comment 9- Cell type (how they have been isolated), cell number per PCR ("generally 10M") and integration assay sensitivity are still obscure.

Response: We added more detail in Methods in revised manuscript.

REVIEWERS' COMMENTS

Reviewer #1 (Remarks to the Author):

The revised manuscript has satisfactorily addressed the concerns noted in the second review.

Reviewer #2 (Remarks to the Author):

Most comments from the previous review have been addressed. Please see remaining comments below

1- The first sentence of the abstract sounds like early ART leads to rapid viral rebound. ATI is followed by viral rebound despite years of treatment

2- Introduction. It is largely admitted that current ART regimens block new cycle of virus replication (with or without integrase inhibitor). Why would the pediatric treatment regimens “permit new or continual proviral seeding”?

3- Results. The new paragraph describing assay validation is poorly written. Supplementary Fig 1A is not referenced in the text and a slope of -2.3 is not good for a standard curve (-3.2?). As mentioned in the previous review, not detecting integration

before 5dpi in pre-stimulated PBMC infected in vitro in absence of RTG treatment is surprising and suggest that the integrated assay might be less sensitive than the total DNA one.

4- As the sensitivity of the integration quantification assay is still unclear, I would suggest removing or rephrasing the following sentences:

- “Our studies showed that in contrast to adults, skewing of viral integration was observed in neonatal macaques...” (Abstract page 2)

- “the onset of integrated reservoir seeding did not appear until day 3 of infection” (Results page 5)

RE: NCOMMS-22-04053 Final, point to point responses to reviewer comments

Reviewer #1:

The revised manuscript has satisfactorily addressed the concerns noted in the second review.

Response: *We appreciate reviewer's positive feedback and valuable comments to improve our manuscript, we also would like to thank the reviewer for taking the time to review the manuscript.*

Reviewer #2

Most comments from the previous review have been addressed. Please see remaining comments below:

1. The first sentence of the abstract sounds like early ART leads to rapid viral rebound. ATI is followed by viral rebound despite years of treatment.

Answer: This sentence has been revised.

2. Introduction. It is largely admitted that current ART regimens block new cycle of virus replication (with or without integrase inhibitor). Why would the pediatric treatment regimens "permit new or continual proviral seeding"?

Answer: Sorry to make confused. Although incorporation of nucleoside reverse transcriptase inhibitors (NRTIs) into the nascent viral DNA terminates its synthesis in the viral life cycle, current early ART regimen with a combination of dual-NRTI (AZT+3TC) plus protease inhibitors (LPV/r) may not fully prevent initial viral RNA/DNA production since total viral RNA/DNA are detected as early as 6h after SIV infection, any residue viral DNA could potentially increase the risk of proviral reservoir seeding. This concept is also supported by off-ART viral rebound observed in infant macaques receiving late ART regimen containing integrase inhibitor. Overall, all these points suggest that proviral reservoirs, once established, are likely difficult to eradicate by conventional cure strategies, and prevention of initial viral integration may be critical to overcome this major obstacle. We have clarified it in the revised version.

3. Results. The new paragraph describing assay validation is poorly written. Supplementary Fig 1A is not referenced in the text and a slope of -2.3 is not good for a standard curve (-3.2?). As mentioned in the previous review, not detecting integration before 5dpi in pre-stimulated PBMC infected in vitro in absence of RTG treatment is surprising and suggest that the integrated assay might be less sensitive than the total DNA one.

Answer: We thoroughly edited the paragraph and referenced Supplementary Fig. 1a in the text. We also corrected the typo and made the graph of standard curve clear in revised Supplementary Fig. 1a. For a geometric efficiency of 100% in qPCR, the slope is -3.32. The slope value here is shallower, implying acceptable amplification efficiency. As for undetectable proviral DNA before 5dpi (not examined at 4dpi) in pre-stimulated infant PBMCs infected *in vitro* in absence of RTG treatment, these results were *de facto* consistent with *in vivo* data that viral genome integration was not observed in PBMCs in neonates within 1-3dpi after SIV infection. Note that there are still tremendous gaps in our understanding of pediatric HIV infection and treatment outcomes thus far. For example, plasma viral load could be detected (or higher) in some neonates as early as one day post SIV infection but undetectable (or lower) in others; all neonatal

animals infected with identical SIV inoculum after birth, including three animals born at same day, only one animal on early treatment at 3dpi shows viral rebound off-ART; some infants infected with HIV/SIV show rapid AIDS development but not others, and so on, which remain elusive. In the qPCR assay, the total and integrated assays were parallelly performed within same batch for each sample, generating comparable data with similar sensitivity. Overall, this result mentioned should make sense, at least not completely question the assay *per se*. We added in the Discussion. Actually, we also primarily tested viral integration using fresh PBMCs from SIV naïve adult animals. Proviral DNA in adult PBMCs without treatment was detected as early as 1dpi while shown individual sample difference: undetectable in one of three cell samples yet detected in another two cell samples (11 and 5.5 copies of proviral DNA per million cells) at 1dpi; detected in all cell samples at 3dpi (74, 38 and 67 copies of proviral DNA per million cells), essentially consistent with the adult animal study that viral rebound off-ART is observed in 20% adult macaques when early treatment is initiated at day 1 post SIV infection, 100% viral rebound off-ART with ART initiation at 3dpi (Whitney, *Nat Commun* 2018).

4. As the sensitivity of the integration quantification assay is still unclear, I would suggest removing or rephrasing the following sentences: “Our studies showed that in contrast to adults, skewing of viral integration was observed in neonatal macaques...” (Abstract page 2); “the onset of integrated reservoir seeding did not appear until day 3 of infection” (Results page 5).

Answer: As we mentioned and discussed in the text and previous rebuttal letter, viral reservoir assay and its sensitivity, especially for proviral reservoirs, are highly challenging for all current scalable assays, absolute quantification is essentially not reachable, and sensitivity of assay is also highly associated with many aspects. With the improved assay here, the treatment outcomes with viral remission basically correlate with status of viral integration (but not other parameters examined), thus trying to interpret the possible mechanisms. As suggested, we rephrased the sentences in the revised text.